# ROYAL SOCIETY
# OPEN SCIENCE

health and disease and epidemiology/behaviour

schools, COVID19, expert judgement, transmission

# A novel approach for evaluating contact patterns and risk mitigation strategies for COVID-19 in English primary schools with application of structured expert judgement

R. S. J. Sparks[1], W. P. Aspinall[1,2], E. Brooks-Pollock[3], R. M. Cooke[4], L. Danon[5], J. Barclay[6], J. H. Scarrow[7] and J. Cox[8]

[1]School of Earth Sciences, University of Bristol, Bristol BS8 1RJ, UK
[2]Aspinall and Associates, Tisbury SP3 6HF, UK
[3]School of Veterinary Sciences, University of Bristol, Office OF24, Churchill Building, Langford, Bristol BS40 5DU, UK
[4]Resources for the Future, 1616 P St NW, Washington, DC 20036, USA
[5]Department of Engineering Mathematics, University of Bristol, Ada Lovelace Building, University Walk, Bristol BS8 1TW, UK
[6]School of Environmental Sciences, University of East Anglia, Norwich Research Park, Norwich NR4 7TJ, UK
[7]Departamento de Mineralogía y Petrología, Facultad de Ciencias, Universidad de Granada, 18071 Granada, Spain
[8]The Royal Society, 6–9 Carlton House Terrace, London SW7 5QR, UK

RSJS, 0000-0001-7173-2899; WPA, 0000-0001-6014-6042;
EB-P, 0000-0002-5984-4932; LD, 0000-0002-7076-1871

**Author for correspondence:**
R. S. J. Sparks
e-mail: steve.sparks@bristol.ac.uk

Personal contacts drive COVID-19 infections. After being closed (23 March 2020) UK primary schools partially re-opened on 1 June 2020 with social distancing and new risk mitigation strategies. We conducted a structured expert elicitation of teachers to quantify primary school contact patterns and how contact rates changed upon re-opening with risk mitigation measures in place. These rates, with uncertainties, were determined using a performance-based algorithm. We report mean number of contacts per day for four cohorts within schools, with associated 90% confidence ranges. Prior to lockdown, younger children (Reception and Year 1) made 15 contacts per day [range 8.35] within school,

older children (Year 6) 18 contacts [range 5.55], teaching staff 25 contacts [range 4.55] and non-classroom staff 11 contacts [range 2.27]. After re-opening, the mean number of contacts was reduced by 53% for young children, 62% for older children, 60% for classroom staff and 64% for other staff. Contacts between teaching and non-teaching staff reduced by 80%. The distributions of contacts per person are asymmetric with heavy tail reflecting a few individuals with high contact numbers. Questions on risk mitigation and supplementary structured interviews elucidated how new measures reduced daily contacts in-school and contribute to infection risk reduction.

## 1. Introduction

In the United Kingdom, multiple non-pharmaceutical interventions have been implemented to control the spread of COVID-19 through reducing the number of contacts between people. In April and May 2020 these included shutting all schools, thus reducing contacts between children, staff and parents, while some provision for children of frontline workers and special needs pupils was maintained. On 11 May 2020 a partial re-opening of schools was announced: selected primary school-age children would return on 1 June in England. The returning cohorts included Reception, Year 1 and Year 6, as well as children of frontline workers and those identified as vulnerable. Nursery age children were also invited to return. Schools fully re-opened in September.

The partial re-opening of primary schools in England was widely debated with concerns from some parents, teaching unions and teacher associations about the safety of the children and school staff. There was also concern about the effect on infection rate in the wider community, for example by triggering a second wave. Some schools re-opened on 1 June, some delayed their re-start, while, in other cases, schools did not re-open under the advice of local education authorities. Data from DfE indicates on 15 July that about 88% of state-funded primary schools in England had re-opened to some extent. The community response was variable and between 1 and 15 June 2020 approximately one-third of eligible children returned. The numbers had increased to 41% (Year 1) and 49% (Year 6) on 2 July. Between 18 May and 31 July 247 COVID-19 related incidents were reported in schools of which 116 were tested positive [1].

Person-to-person contact patterns, as drivers of transmission for close-contact infections, are an essential component of epidemiological models [2–6]. There is, however, a paucity of empirical information about the contact patterns of younger children, especially in school settings, due to the challenges involved in collecting data from children. Children are rarely included in direct surveys. However, studies of contact patterns in US elementary and a French primary school using mote devices provide datasets that are informative within those specific contexts [7,8].

Here, we address this fundamental data gap by applying structured expert judgement (SEJ) using leaders in English primary schools as experts. SEJ is a well-established approach to quantifying parameters and their attendant uncertainties where there are no data, the data are sparse or of poor quality, are highly empirical in character or have large associated uncertainties [9,10]. This is the case for many epidemiological parameters, and especially contact patterns in children. SEJ has been widely applied to risk assessment and uncertainty analysis in many areas of science, engineering, the environment, business and public health [10]. The classical model (CM) for SEJ has been deployed in several public health policy applications [11–17].

We identified 33 volunteers from within the Royal Society Schools Network. This group of experts are leaders and senior staff of largely state-funded primary schools in England. The focus of our SEJ was to use these volunteers and their experience to quantify contact rates and patterns within schools and investigate how these had changed between 'pre-COVID' and 'new normal' times.

We also took the opportunity to survey school staff about their risk mitigation strategies and asked for information to assess their effectiveness. Under current circumstances, the usual approach to SEJ involving typically 1 or 2 days of face-to-face discussions between experts was impractical and would in any case contravene social distancing requirements. We, therefore, undertook the SEJ elicitation process by email dialogues with the volunteers and, to improve our understanding of the responses, we augmented the elicitation procedure with structured interviews of six of the participants.

## 2. Methods and data

To determine variable or parameter uncertainty distributions we used a validated elicitation method, the CM for SEJ. The method is described in [9,10]. The distinctive feature of the CM is that experts, in a group

being elicited, are empirically calibrated for their ability to judge uncertainties in terms of statistical accuracy taken over a set of knowable test variables from within the subject matter domain. The resulting performance-based calibration scores are used as weights to create a synthetic combined distribution which is also scored for statistical accuracy and informativeness. The so-called decision maker (DM) is in effect, a synthesized pseudo-expert, which represents the group's collective judgement. Expert weights are predicated on the mathematical concept of 'proper scoring' rules, with the resulting DM solutions providing a rational consensus of the experts' judgements, weighted according to the individual calibration performance scores. From here in the article, the volunteer primary school staff will be termed experts.

Calibration questions, to which true answers are known *post hoc*, for the SEJ were prepared (see electronic supplementary material, S1, §1). Elicitation questions were devised by the authors to elucidate contact patterns within schools (electronic supplementary material, S1, §1). We also took the opportunity to ask the experts about risk mitigation measures that they have put in place (responses are given in electronic supplementary material, S1, §3).

During national lockdown and with experts distributed in primary schools across England, the normal procedure was not possible. A briefing session was held by zoom where 17 of the experts participated. The briefing session was recorded, giving those who could not attend the chance to hear the proceedings. For both the seed and elicitation questions, Aspinall and Sparks were available to respond to queries and clarifications by email. To compensate for the lack of a formal meeting, six teachers were chosen for structured interview [18]. The transcripts of these interviews are provided in electronic supplementary material, S1 (§2). The questionnaire protocol ensured participants described the thinking behind their quantitative answers but also allowed a free exploration of topics the teachers perceived of relevance to adult and child contacts.

## 2.1. Data resources

We extracted national data on primary schools in England from DfE links for 2019 and 2018.[1] There are 16 769 state-funded primary schools in England with 4 727 090 pupils, 216 500 teachers, 176 679 teaching assistants and 132 085 other support staff. Daily national attendance of pupils, teachers, teaching assistants and ancillary staff in schools was also accessed.[2]

## 2.2. The study schools

The major source of our data comes from the expert elicitation of a number of volunteers from primary schools in England who responded to a call to participate as experts able to characterize contact rates between children, teachers and other staff in schools from their extensive professional experience. The volunteers were drawn from the Royal Society Schools Network, consisting of 900 schools across the four nations of the UK; the network includes 1300 teachers. The network is open to all schools, with 86% of schools being state-funded and the rest independents. However, all but one of the 34 volunteer schools were state schools. The focus of the network is STEM education and so the majority of our recruited teachers had STEM backgrounds through a science degree. However, a few were teachers with arts-based degrees, but with many years of experience teaching STEM subjects. While there is a slight tendency for our volunteer schools to be in regions where students have greater rates of access to higher education, there is no compelling evidence that they represent only high performing institutions.

The STEM background of the volunteers was an advantage for eliciting numerical data framed by basic statistical concepts with which the teachers were already familiar. While 34 volunteers contributed responses to the first questionnaire, seven did not complete the calibration questions fully. Thus, we had 27 persons who both completed the calibration questions and whose judgements fulfilled the requirements for performance-weighted pooling in the first round.

The primary schools ranged in size from 65 pupils to 910 with an average of 376 children (cf. the national average of 282 pupils). The schools are geographically well distributed: Eastern England (4);

---

[1]https://assets.publishing.service.gov.uk/government/uploads/system/uploads/attachment_data/file/826252/Schools_Pupils_and_their_Characteristics_2019_Accompanying_Tables.xlsx; https://www.besa.org.uk/key-uk-education-statistics/; https://www.gov.uk/government/statistics/school-workforce-in-england-november-2018.

[2]https://www.gov.uk/government/collections/attendance-in-education-and-early-years-settings-during-the-coronavirus-covid-19-outbreak.

Southwest (8); London (5); Northwest (5); Midlands (5); South and Southeast (6); Northeast (1) and were a mixture of urban and rural settings. An indicator of socioeconomic setting is provided by the POLAR4 classification based on the likelihood of children participating in higher education. Schools are divided into quintiles from quintile 1 (least likely) to quintile 5 (most likely). Our study schools are fairly evenly distributed across the quantiles but with a slight bias towards higher-achieving catchments: quintile 1 (4), quintile 2 (4), quintile 3 (7), quintile 4 (7) and quintile 5 (9).

Teaching staff ranged from 53 to 5 teachers (average 18) and support staff ranged from 66 to 4 (average 24). The national average number of teachers and support staff per primary school are 13 and 18, respectively. The average school size is based on pupils not teachers and they are 30% greater than the national average. They have almost identical average pupil/teacher ratios (20.2) to the national average ratio of 21.8. All the recruited experts were in positions of senior management or authority with descriptions including: head or deputy head teachers (21); head of department or subject coordinator (8); regional or area mentor (5). With the exception of the mentors, all the experts had hands-on classroom experience and roles.

We divided the people in the schools into four cohorts based on the children's year groups and staff roles. While a simplification, the cohorts are expected to have different contact characteristics, e.g. with respect to interactions between children, between children and adults and between adults. The cohorts are as follows: *Cohort 1* are Nursery, Reception and Year 1 children; *Cohort 2* are Year 2–6 children, noting that Year 2–5 children are those of key workers and from vulnerable environments; *Cohort 3* are classroom teachers and teaching assistants; *Cohort 4* are non-teaching staff such as administrators, cooks, etc., some of whom are expected to have more limited contact with children.

## 2.3. Elicitation methods

Our expert elicitation was conducted under a protocol of confidentiality and non-attribution in order to encourage individual participants to express their own professional judgements about contact patterns, and to remove any constraint, such as expressing only official or policy expectations.

### 2.3.1. The questions

Table 1 lists the elicitation questions relating to contact numbers and table 2 lists questions related to risk mitigation. The full questions and data for completed responses are provided in electronic supplementary material, S1 (§1).

Because epidemiological models use contact data as a basis for modelling transmission of infection [5,7,19], the majority of our elicitation questions focus on contacts between persons in schools. The greater the number of contacts and the longer the duration of those contacts, the greater the chance of infection transmission [19]. In this study, a contact is defined as a conversation or interaction at a spacing of 1 m or less for 5 min or more. Previous studies of contacts in elementary/primary schools [7,8], using personal position wireless mote devices, found that the relationship between contact numbers and duration exhibits a power-law distribution. Those studies, acknowledging the arbitrariness, had chosen contacts of 5 min or more as significant, to be counted towards daily contact totals. However, mote devices can detect contacts at distances of up to 3 m depending on local transmission paths affecting signal strength between devices. Thus, our definition of a contact is, on average, generally comparable with results from these previous studies. We also justify this choice *a posteriori* by showing the contribution of short contacts only add modestly to total contact duration, which is dominated by long-duration contacts.

We asked two kinds of questions to characterize daily contacts for individuals within a cohort. One kind of question aimed at estimating daily contact counts for a *typical representative* individual on a normal pre-COVID day (Q2a, Q5a, Q8a and Q10a) and then in 'new normal' times (Q3a, Q6a, Q9a and Q11a). The other kind of question aimed at estimating variations between different individuals in terms of their daily contacts on a pre-COVID day (Q2b, Q5b, Q8b and Q10b) and in new normal times (Q3b, Q6b, Q9b and Q11b). In these latter questions, experts are asked to think about the variation of contacts for different individuals within the cohort. Here 'least' and 'most' are used to make comparison with the median values elicited in the first kind of the question for the 'typical' individual. Combining the results for the 50th percentile value from questions of the first kind with values of 'least' and 'more' from questions of the second kind provides a measure of the variability of contacts for individuals in a cohort; this represents a quantitative realization with meaning for epidemiological modelling.

**Table 1.** Quantitative elicitation questions related to contacts (see §1 of electronic supplementary material, S1 for full questionnaire). The Table format is as sent to the experts with some words in bold or underlined to give emphasis to help with the clarity if the questions.

| number | question |
| --- | --- |
| 1b | *If you use bubbles please describe the number of pupils in a bubble and the approximate spacing between bubbles during class time. The opportunity is given to give separate answers for Cohort 1 and Cohort 2.* |
| 2a | *How many people does a typical[a] Cohort 1 child come into face-to-face contact with (conversation within 1 m for 5 min or more) on a normal school day in a COVID-free world?* |
| 2b | *Thinking about the behaviour of individual children,[b] give a range of contacts for Cohort 1 children around your central answer to Q2a for normal times. Provide an estimate of the least and most number of contacts for individuals.* |
| 3a | *How many people does a typical Cohort 1 child come into face-to-face contact with (conversation within 1 m for 5 min or more) on a new normal school day?* |
| 3b | *Thinking about the behaviour of individual children give a range of contacts for Cohort 1 children around your central answer to Q3a for new normal times, provide an estimate of the least and most number of contacts for individuals.* |
| 4 | *Do you think that there is any significant difference[c] in the contacts made by nursery, reception and Year 1 children? If yes, please indicate the likely difference in contact referenced to reception age children. Put a percentage to indicate a difference (either way); for example you might judge that Year 1 children have 80% of the contacts of a reception child.* |
| 5a | *How many people does a typical Cohort 2 child come into face-to-face contact with (conversation within 1 m for 5 min or more) on a normal school day?* |
| 5b | *Thinking about the behaviour of individual children, give a range of contacts for Cohort 2 children around your central answer to Q5a for normal times. Provide an estimate of the least and most number of contacts for individuals.* |
| 6a | *How many people does a typical Cohort 2 child come into face-to-face contact with (conversation within 1 m) on a new normal school day?* |
| 6b | *Thinking about the behaviour of individual children, give a range of contacts for Cohort 2 children around your central answer to Q6a for new normal times. Provide an estimate of the least and most number of contacts for individuals.* |
| 7 | *Do you think that there is any difference in the contacts made by Year 2–5 children compared to Year 6.[d] If yes, please indicate the likely difference in contact referenced to Year 6 age children. Put a percentage to indicate a difference (either way); for example you might judge that Year 2–5 children have 120% or 80% of the contacts of a Year 6 child.* |
| 8a | *How many people (both children and other adults) does a Cohort 3 adult come into face-to-face contact (within 1 m for 5 min or more) with on a normal school day?* |
| 8b | *Thinking about the behaviour of individual Cohort 3 adults, give a range of contacts for Cohort 3 adults around your central answer to Q8a for normal times. Provide an estimate of the least and most number of contacts for individuals.* |
| 9a | *How many people (both children and other adults) does a Cohort 3 adult come into contact (within 1 m for 5 min or more) with on a new normal school day?* |
| 9b | *Thinking about the behaviour of individual Cohort 3 adults, give a range of contacts for Cohort 3 adults around your central answer to Q9a for new normal times. Provide an estimate of the least and most number of contacts for individuals.* |
| 10a | *How many people (both adults and children) does a Cohort 4 adult come into face-to-face contact (within 1 m for 5 min or more) with on a normal school day?* |
| 10b | *Thinking about the behaviour of individual Cohort 4 adults, give a range of contacts around your central answer to Q10a for normal times. Provide an estimate of the least and most number of contacts for individuals.* |

(Continued.)

**Table 1.** (*Continued.*)

| number | question |
|---|---|
| 11a | *How many people (both adults and children) does a Cohort 4 adult come into face-to-face contact (within 1 m for 5 min or more) with on a <u>new normal</u> school day?* |
| 11b | *Thinking about the behaviour of individual Cohort 4 adults, give a range of contacts around your central answer to Q16 for <u>new normal</u> times. Provide an estimate of the least and most number of contacts for individuals.* |
| 12 | *How many children does **an adult (from both Cohort 3 and 4)** have physical/face-to-face contact (within 1 m for 5 min or more) with during a typical day, in <u>normal</u> times?* |
| 13 | *How many adults does **an adult (from both Cohort 3 and 4)** have physical/face-to-face contact (within 1 m for 5 min or more) with during a typical day, in <u>normal</u> times?* |
| 14 | *How many children does **an adult** have physical/face-to-face contact (within 1 m for 5 min or more) with during a typical day, in <u>new normal</u> times?* |
| 15 | *How many adults does **an adult** have physical/face-to-face contact (within 1 m for 5 min or more) with during a typical day, in <u>new normal</u> times?* |

[a]In this and subsequent similar questions you are being asked about the typical or average behaviour and your uncertainty in this average or typical behaviour.

[b]This question and subsequent similar questions you are asked to think about the extreme behaviours of individuals in your school.

[c] It seems likely to us that there will be little difference in the behaviour and management regime for very young children but this is an opportunity for you to disagree.

[d]Numbers, character and organization of Year 2 to 5 (emergency workers and vulnerable children) may differ from Year 6 affecting contacts. This question might not be relevant to some schools.

**Table 2.** Questions related to risk mitigation.

| question | |
|---|---|
| 1a | *Please describe your strategy to reduce close contacts between pupils (about 50 words). You might want to discriminate between Cohort 1 and 2 above.* |
| 16 | *Adherence. All schools will seek to provide a safe environment following guidelines (e.g. DfE) where feasible and from their own management decisions. Of course some recommendations and procedures may be easier to adhere to than others. We would like your assessment of the extent to which your school is able to follow guidelines. 100% means perfect adherence while 0% means no adherence at all. We have left some space below if you want to make any comments on any factors that make it difficult to reach 100%.* |
| 18 | *Describe briefly the cleaning regime in your school using disinfectant (maximum 50 words).* |
| 19 | *Describe the policy for parents to drop off and pick up their children each day (50 words maximum).* |
| 20 | *Weather might affect the ability to socially distance and being outdoors is recognized as being less risky than indoors. Estimate the increase in number of contacts as a percentage increase among children as a consequence of bad weather (e.g. an answer of 30% will indicate the number of contacts increase by this percentage due to play breaks being inside).* |
| 21 | *How many times a day do your children wash their hands? Give a range if applicable.* |
| 22 | *What is your policy if a child or adult staff develops possible COVID19 symptoms (either outside school or during school hours) (50 words maximum)?* |
| 23 | *What is your policy if a parent or another relative of a child contracts COVID19?* |
| 24 | *Other comments. Please make any remarks about additional risk factors that you think should be considered (maximum 100 words).* |

By contrast, Questions Q12–Q15 concerned contacts between individuals who are members of different cohorts. These questions were augmented by questions on bubble sizes (Q1b) and on contact rate comparisons between children of different age (Q4 and Q7). Most questions required elicitation of a range to reflect epistemic and aleatory uncertainties, which are discussed in more detail in the Results sections below. Questions were also asked on risk mitigation measures (Q1a, Q16–Q24) which required a mixture of quantitative and qualitative responses.

Before circulating the calibration and elicitation questions to the teachers, the questions were reviewed by Prof. Andrew Noyes (Faculty of Education, University of Nottingham) to check their clarity. Some revisions were made as a consequence.

Normally, experts are convened in a plenary meeting or workshop, in order to conduct the elicitation in a facilitated, structured manner. Such a meeting usually covers: introduction to the methodology; calibration of the experts using seed questions; presentation and discussion of the questions; and a time for the experts to answer the questions. Depending on the scope, complexity and number of issues to be elicited, a meeting might typically last from one to three days, with time allowed for discussion on the evidence that can inform individual experts' responses to the questions. After the experts' responses have been processed, it is customary to discuss the findings with the participating group; usually, there is an opportunity for participating experts to critique results and, where necessary, for some questions to be clarified and any disputed critical items re-elicited.

### 2.3.2. The participants

The elicitation produced 27 complete responses from teachers who had undertaken the CM calibration process. Thus, each had a personal statistical accuracy score and an information score (table 3). These scores act as a relative performance weight for each participant that is used when pooling their judgements jointly and enumerating uncertainty for specific target/query items. CM non-optimized item weights combination solutions (i.e. all experts given some weight based on calibration scores) are listed in table 3. These weights are not the same as ascribing equal weights to all participants. When combining judgements collectively, each person's uncertainty distribution is weighted according as their performance score, so each contributes with some real positive weight to the overall outcome.

The calibration score is calculated on the basis of a $\chi^2$-test comparing the empirical and theoretical distributions for realizations falling in the expert's interquantile intervals for the calibration questions as a set. Its employment here is motivated by the theory of proper scoring rules [9] rather than simple hypothesis testing and the resulting $\chi^2$ $p$-values are used to measure the experts' statistical accuracies. Using Shannon's relative information statistic, the $\chi^2$-test takes the frequencies with which the expert's assessed calibration variable values fall within various ranges, and compares these with the counts of actual (known) item values in the same ranges; this produces relative probabilities of the match per item, which can be summed over all calibration items to form a measure of the expert's statistical accuracy. When combined with the expert's information metric [9] in a product, this $p$-value provides the 'statistical accuracy' part of the expert's performance score. In effect, a calibration $p$-value can be thought equivalent to the probability that an expert's performance would be regarded as statistically accurate by chance but, in the CM, it is not used in the sense of a hypothesis test. Rather, it is the formal mathematical basis for enumerating the metric for the statistical accuracy of the expert's assessments and is used, with a companion information metric, for scoring performance in assessing uncertainties. A high $p$-value indicates a close correspondence between the expert's assessment values and the known values (a perfect score would be $p = 1$); a very low $p$-value signals major deviations exist between the assessed and actual values. The calibration metric is a 'fast' function and typically changes markedly between experts in an elicitation.

# 3. Results

## 3.1. The elicitation

We found that teachers' elicitation does not exhibit any substantive shortcomings compared to other cases, despite being conducted with minimal briefing and without the benefit of a plenary workshop to help focus judgements. In short, the teachers' performances proved to be as strong, collectively, as those of many other groups of experts (see [10] for other case profiles). The calibration and weighting profile of the group is very similar to that of other professional expert elicitations; in figure 1 a profile

**Table 3.** Elicitation experts' calibration.

| expert | statistical accuracy $p$-value | relative information metric | no. seed items | relative performance score | normalized weight with DM |
|---|---|---|---|---|---|
| E01 | $8.92 \times 10^{-3}$ | 2.342 | 7 | $1.29 \times 10^{-2}$ | $1.55 \times 10^{-2}$ |
| E02 | $1.60 \times 10^{-3}$ | 2.262 | 7 | $2.83 \times 10^{-3}$ | $3.39 \times 10^{-3}$ |
| E04 | $7.07 \times 10^{-4}$ | 2.597 | 7 | $1.00 \times 10^{-3}$ | $1.20 \times 10^{-3}$ |
| E05 | $1.89 \times 10^{-4}$ | 2.334 | 7 | $2.79 \times 10^{-4}$ | $3.35 \times 10^{-4}$ |
| E07 | $1.62 \times 10^{-2}$ | 2.328 | 7 | $1.95 \times 10^{-2}$ | $2.34 \times 10^{-2}$ |
| E08 | $1.24 \times 10^{-2}$ | 2.317 | 7 | $2.21 \times 10^{-2}$ | $2.65 \times 10^{-2}$ |
| E09 | $1.82 \times 10^{-1}$ | 1.574 | 7 | $1.79 \times 10^{-1}$ | $2.15 \times 10^{-1}$ |
| E10 | $1.82 \times 10^{-1}$ | 2.204 | 7 | $9.21 \times 10^{-2}$ | $1.10 \times 10^{-1}$ |
| E11 | $1.24 \times 10^{-2}$ | 2.331 | 7 | $1.20 \times 10^{-2}$ | $1.44 \times 10^{-2}$ |
| E12 | $4.13 \times 10^{-9}$ | 2.639 | 7 | $1.27 \times 10^{-8}$ | $1.52 \times 10^{-8}$ |
| E13 | $2.45 \times 10^{-7}$ | 2.060 | 7 | $3.78 \times 10^{-7}$ | $4.53 \times 10^{-7}$ |
| E17 | $7.07 \times 10^{-4}$ | 1.622 | 7 | $9.07 \times 10^{-4}$ | $1.09 \times 10^{-3}$ |
| E18 | $1.62 \times 10^{-2}$ | 2.239 | 7 | $2.01 \times 10^{-2}$ | $2.41 \times 10^{-2}$ |
| E19 | $7.07 \times 10^{-4}$ | 2.410 | 7 | $1.36 \times 10^{-3}$ | $1.63 \times 10^{-3}$ |
| E20 | $1.24 \times 10^{-2}$ | 2.220 | 7 | $2.45 \times 10^{-2}$ | $2.94 \times 10^{-2}$ |
| E21 | $2.01 \times 10^{-2}$ | 2.231 | 7 | $2.83 \times 10^{-2}$ | $3.40 \times 10^{-2}$ |
| E22 | $7.07 \times 10^{-4}$ | 1.398 | 7 | $7.92 \times 10^{-4}$ | $9.50 \times 10^{-4}$ |
| E23 | $6.52 \times 10^{-5}$ | 2.050 | 7 | $8.92 \times 10^{-5}$ | $1.07 \times 10^{-4}$ |
| E24 | $1.24 \times 10^{-2}$ | 2.383 | 7 | $1.47 \times 10^{-2}$ | $1.76 \times 10^{-2}$ |
| E25 | $1.89 \times 10^{-4}$ | 1.650 | 7 | $3.06 \times 10^{-4}$ | $3.67 \times 10^{-4}$ |
| E26 | $2.01 \times 10^{-2}$ | 2.133 | 7 | $2.28 \times 10^{-2}$ | $2.74 \times 10^{-2}$ |
| E28 | $1.62 \times 10^{-2}$ | 1.944 | 7 | $2.22 \times 10^{-2}$ | $2.67 \times 10^{-2}$ |
| E30 | $1.42 \times 10^{-1}$ | 2.184 | 7 | $9.95 \times 10^{-2}$ | $1.19 \times 10^{-1}$ |
| E32 | $2.01 \times 10^{-2}$ | 2.634 | 7 | $1.89 \times 10^{-2}$ | $2.27 \times 10^{-2}$ |
| E33 | $1.52 \times 10^{-3}$ | 2.221 | 7 | $1.74 \times 10^{-3}$ | $2.09 \times 10^{-3}$ |
| E35 | $1.52 \times 10^{-3}$ | 2.322 | 7 | $2.55 \times 10^{-3}$ | $3.06 \times 10^{-3}$ |
| E36 | $1.89 \times 10^{-4}$ | 2.176 | 7 | $1.47 \times 10^{-4}$ | $1.77 \times 10^{-4}$ |
| *solution DM* | *$5.33 \times 10^{-1}$* | *0.813* | | *$2.33 \times 10^{-1}$* | *$2.79 \times 10^{-1}$* |

is shown for a medical panel with the same number of participants (N.B. some points overlap). While the overall range of the teachers' statistical accuracy $p$-values—from highest to lowest—is not atypical, in this case, there are fewer individuals with very low $p$-values (i.e. $p < 10^{-6}$) than in the medical panel; the latter represents the more usual case. With the exception of one teacher (a very low $p$-value outlier not shown here), the majority of the teachers' $p$-values are more clustered at higher $p$-values than those obtained for the medical panel. On the other hand, the relative information scores of the teachers are generally lower than those of the medical experts; this may reflect inherent differences in the natures of the precision of the data types each group were considering.

The combination DM—the combination of all experts—is statistically more accurate than any individual expert. This finding is commonplace with elicitations processed with the CM. Colson & Cooke [10] report (their table 1) that best expert statistical accuracy is superior to the statistical accuracy of the corresponding performance-weights DM solutions (global or item) in only one SEJ study out of 32. The range of relative information metric scores is also typical; this is a 'slow' function that does not vary greatly from one expert to another. Higher values indicate experts who provided tighter (more informative) uncertainty ranges. However, there is here, and usually, an inverse relationship between informativeness and statistical accuracy: experts who are too narrow or too precise with their uncertainty judgements tend to 'miss the target' too frequently and their performance scores are reduced, as a consequence. On the other

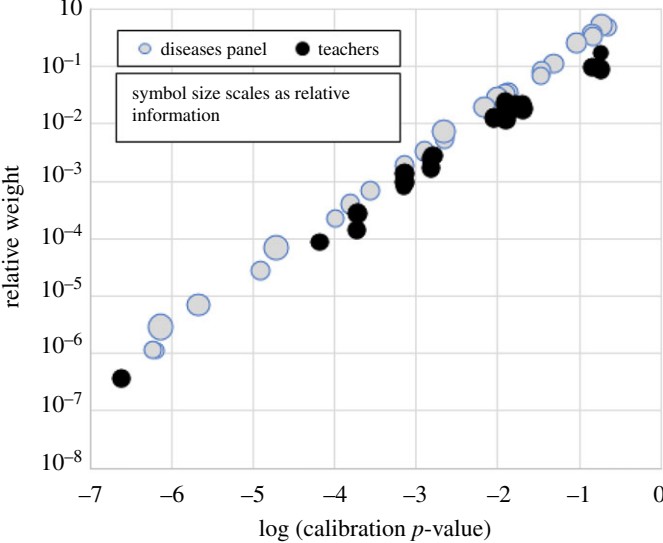

**Figure 1.** Teachers' group calibration and weights profile, compared with a childhood diseases medical experts panel of the same size. Each panel comprised 27 participants; the two panels were scored on different subject-matter calibration questions. The plot shows individuals with calibration $p$-value at least $10^{-7}$, and symbol size scales as the expert's relative information score; there are several overlapping points. One teacher from the schools study group with the lowest $p$-value was removed just to make the comparison balanced in terms of participating numbers.

hand, the goal is to identify those experts who are both informative *and* statistically accurate. This is usually a minority of a group, and such persons are not identifiable *a priori* on common grounds such as professional standing; the best uncertainty assessors are discovered only after calibration. The DM has a low information score compared to individual experts, and this is the price paid for the DM's superior statistical accuracy. As noted above, it is well established that individual experts tend to underestimate true variable uncertainties [20] and pooling via the DM redresses this trait. The combination DM out-performs the best single expert and provides a set of target item solutions that is superior to any single expert.

Individual expert's relative performance scores (column 5 in table 3) are computed from the products of their $p$-value and information scores, per item, and are un-normalized. When the DM is included as a synthetic expert, performance scores are adjusted and normalized to sum to unity across the group (with DM included). These are the relative weights (column 6 in table 3) that are used in the CM for combining judgements on target items. Two experts (E09 and E10) jointly achieve the best statistical accuracy scores (0.182); their judgement influences are differentiated in the analysis by their information scores—one is more informative than the other and is consequently rewarded with a marginally higher overall weight (see two rightmost black points on figure 1).

Prior to the main elicitation and before 1 June we asked the experts to forecast the proportion of returning pupils and teachers on two dates (1 and 15 June). When completing these forecasts some teachers indicated that their estimates and ranges were based on surveys of parents before 1 June, conducted for planning purposes. As such, these percentages did not represent personal judgements. However, without comments from other teachers, it was not possible to know how general pre-return surveys were, or how many respondents had provided percentages data based on similar surveys. Thus, for the purposes and goal of our elicitation exercise, we chose to regard all the inputs on the percentages of pupils returning as representing objective, informed judgements and treated them uniformly when processing the group's responses.

The forecasts were received before or by 27 May 2020. There was a wide range of responses (figure 2 and table 4), indicating a remarkably diverse set of circumstances in individual schools and community enthusiasm or lack of enthusiasm for a return to school. This was borne out by the expert interviews where the actual return had been highly variable across just six schools driven both by community perception and school's mitigation measures. However, when averaged over all of the teachers, the median forecasts are very close (indeed, for pupil's attendance, identical) to the national attendance on 1 June (table 4). The national attendance on 15 June was similar to 1 June but had increased to levels similar to those anticipated by the teachers for 15 June by 2 July. These results demonstrate the ability of the teachers to make good forecasts and strengthen the belief that the study schools are a dependable representative sample of primary schools in England.

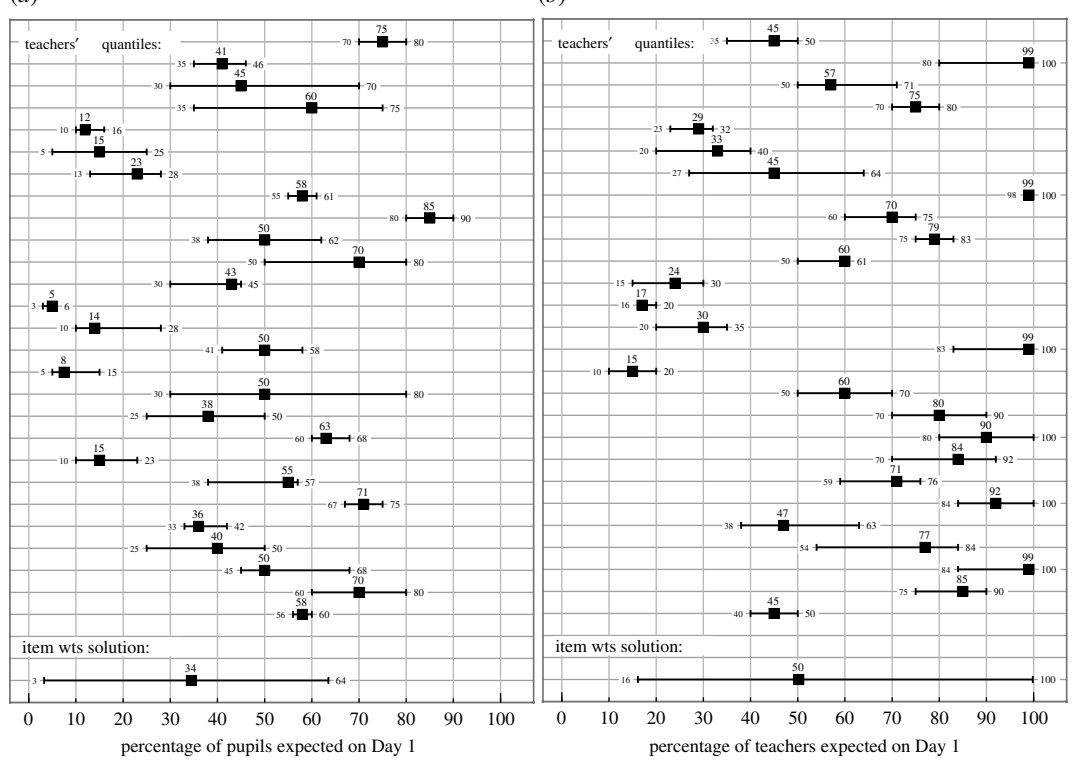

**Figure 2.** Forecasts of number of pupils (*a*) and number of teachers (*b*) returning to school on 1 June 2020. Judgements, expressed as 5th, 50th and 95th percentiles, were received from 27 participating teachers before or by 24 May 2020 (displayed here anonymously and in order received for processing). The bottom row in each panel shows the item weights combination solution for the question (see text).

**Table 4.** Forecasts by expert group of percentages of different persons returning to school on given dates. Numbers for pupils are overall totals for Reception, Year 1 and Year 6 children.

| forecast | 5th percentile | 50th percentile | 95th percentile | school outcome | national outcome |
|---|---|---|---|---|---|
| pupils (1 June) | 3.3 | 34.5 | 63.5 | 40.1 | 34.4 |
| teachers (1 June) | 16.1 | 50.2 | 99.8 | 57.6 | 40.0 |
| support (1 June) | 1.6 | 35.4 | 79.6 | 51.3 | not known |
| pupils (15 June) | 5.9 | 47.8 | 83.6 | no data | 45.3[a] |
| teachers (15 June) | 26.0 | 49.5 | 99.1 | no data | not known |
| support (15 June) | 22.3 | 50.1 | 81.5 | no data | not known |

[a]Data refer to 2nd July.

The six structured interviews strengthened the assessment of the teachers as thoughtful and careful in thinking through the responses to the questions even though they were regarded as challenging. All thought about variations between different children, differences in the school day, differences between class time and breaks, and in roles and interactions of staff. Two of the interviewees had consulted with other staff to help them think about contacts.

## 3.2. Elicitation data

### 3.2.1. Characteristics and uncertainties

We list the results in table 5 in terms of quantiles for the elicitation of a single central estimate. Figure 3 displays data for Cohort 1 illustrating the marked reduction in contacts between pre-COVID and new

**Table 5.** Results of elicitation. Unless otherwise indicated the data are contacts per day.

| question | sub-question | mean | 5th percentile | 50th percentile | 95th percentile |
|---|---|---|---|---|---|
| 1b | Cohort 1. Number per bubble[a] | 11 | 7 | 11 | 16 |
| | Cohort 2. Number per bubble[a] | 14 | 10 | 13 | 15 |
| 2a | Cohort 1. Number face-to-face daily contacts in pre-COVID times | 25 | 7 | 13 | 75 |
| 2b | Cohort 1. Most number of daily contacts in pre-COVID times | 27 | 13 | 40 | 137 |
| | Cohort 1. Least number of daily contacts in pre-COVID times | 13 | 3 | 10 | 41 |
| 2c | **Contact distribution for Cohort 1 in pre-COVID times** | 15 | 8 | 11 | 35 |
| 3a | Cohort 1. Number face-to-face daily contacts in new normal times | 14 | 2 | 8 | 34 |
| 3b | Cohort 1. Most number of daily contacts in new normal times | 11 | 6 | 13 | 50 |
| | Cohort 1. Least number of daily contacts in new normal times | 5 | 1 | 4 | 16 |
| 3c | **Contact distribution for Cohort 1 in new normal times** | 8 | 2 | 5 | 19 |
| 4 | Cohort 1. Daily contacts difference: Nursery relative to Reception | 70% | 0% | 100% | 126% |
| | Cohort 1. Daily contacts difference: Year 1 relative to Reception | 66% | 49% | 70% | 101% |
| 5a | Cohort 2. Number face-to-face daily contacts in pre-COVID times | 25 | 4 | 18 | 70 |
| 5b | Cohort 2. Most number of daily contacts in pre-COVID times | 30 | 19 | 42 | 119 |
| | Cohort 2. Least number of daily contacts in pre-COVID times | 8 | 3 | 9 | 33 |
| 5c | **Contact distribution for Cohort 2 in pre-COVID times** | 18 | 5 | 13 | 55 |
| 6a | Cohort 2. Number face-to-face daily contacts in new normal times | 9 | 1 | 5 | 23 |
| 6b | Cohort 2. Most number of daily contacts in new normal times | 11 | 5 | 9 | 35 |
| | Cohort 2. Least number of daily contacts in new normal times | 4 | 1 | 3 | 11 |
| 6c | **Contact distribution for Cohort 2 in new normal times** | 7 | 1 | 5 | 19 |
| 7 | Cohort 2. Daily contacts difference: Year 2–5 vs Year 6 | 104% | 99% | 100% | 200% |
| 8a | Cohort 3. Number face-to-face daily contacts in pre-COVID times | 40 | 4 | 26 | 147 |
| 8b | Cohort 3. Most number of daily contacts in pre-COVID times | 28 | 5 | 49 | 300 |
| | Cohort 3. Least number of daily contacts in pre-COVID times | 9 | 1 | 10 | 105 |
| 8c | **Contact distribution for Cohort 3 in pre-COVID times** | 25 | 4 | 21 | 55 |
| 9a | Cohort 3. Number face-to-face daily contacts in new normal times | 14 | 0 | 8 | 30 |
| 9b | Cohort 3. Most number of daily contacts in new normal times | 17 | 4 | 15 | 109 |
| | Cohort 3. Least number of daily contacts in new normal day | 3 | 0 | 2 | 15 |
| 9c | **Contact distribution for Cohort 3 in new normal times** | 10 | 1 | 7 | 26 |
| 10a | Cohort 4. Number face-to-face daily contacts in pre-COVID times | 19 | 2 | 11 | 49 |
| 10b | Cohort 4. Most number of daily contacts in pre-COVID times | 20 | 13 | 15 | 105 |
| | Cohort 4. Least number of daily contacts in pre-COVID times | 3 | 1 | 3 | 25 |
| 10c | **Contact distribution for Cohort 4 in normal times** | 11 | 2 | 6 | 27 |
| 11a | Cohort 4. Number face-to-face daily contacts in new normal times | 7 | 0 | 3 | 15 |
| 11b | Cohort 4. Most number of daily contacts in new normal times | 6 | 1 | 4 | 57 |
| | Cohort 4. Least number of daily contacts in new normal times | 2 | 0 | 1 | 19 |
| 11c | **Contact distribution for Cohort 4 in new normal times** | 4 | 1 | 2 | 11 |
| 12 | Cohorts 3 and 4. Number of child daily contacts in pre-COVID times | 20 | 1 | 17 | 84 |
| 13 | Cohorts 3 and 4. Number of adult daily contacts in pre-COVID times | 12 | 2 | 10 | 34 |
| 14 | Cohorts 3 and 4. Number of child daily contacts in new normal times | 8 | 0 | 7 | 25 |
| 15 | Cohorts 3 and 4. Number of adult daily contacts in new normal times | 3 | 0 | 2 | 9 |
| 16 | % school adherence in typical primary school. | 83% | 58% | 95% | 97% |
| 20 | % contacts increase due to weather | +10% | −0% | +1% | +45% |
| 21 | Number hand washes per day by children | 7 | 3 | 7 | 13 |

[a]Denotes represent 5th, 50th and 95th percentile variance spreads on the single values, rather than the usual elicited uncertainty ranges per expert.

Contact distributions are shown in bold as these are judged to be the most appropriate for epidemiological transmission modelling.

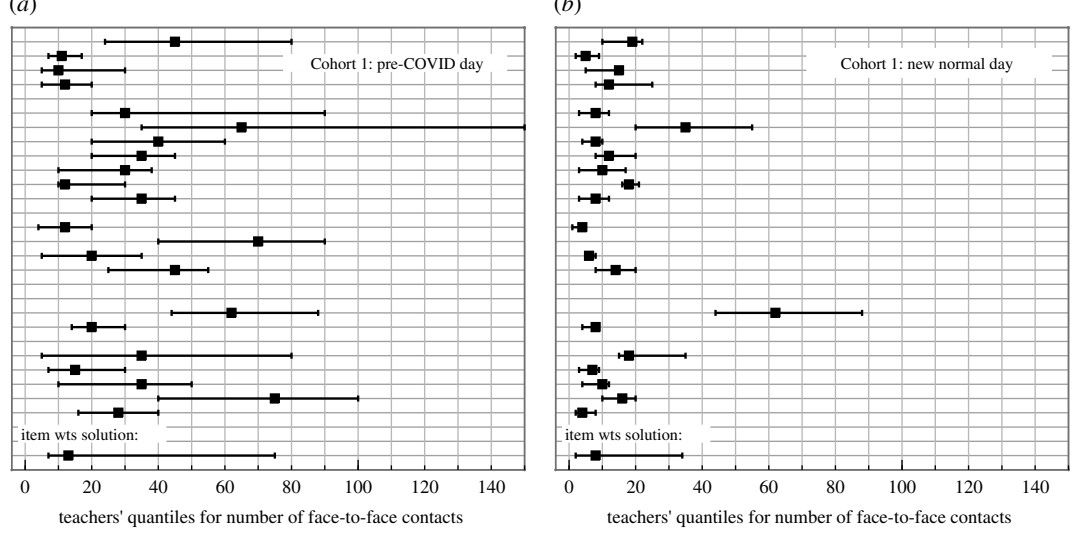

**Figure 3.** Uncertainty range graphs from teachers' quantiles for: (*a*) number of face-to-face contacts in Cohort 1 children on a typical pre-COVID day, and (*b*) the same for a new normal day. These evaluations are based solely on the responses to elicitation questions Q2a and Q3a for the 'typical' individual (see text). The bottom row in each panel shows the item weights combination solution for that particular question.

normal times and the wide range of values given by individual teachers. The wide variations in central estimates and ranges, between teachers, and the occasional outlier judgement are typical of the responses to our contacts questions and, indeed, whenever epistemic uncertainties are large due to paucity of empirical data, of SEJ results in general [10].

Figure 4 shows the range graphs of the performance-weights DM solutions for the daily mean number of contacts for the four cohorts. Substantial reductions in all contact rates from pre-COVID to new normal times are evident. Figure 5 shows an example of the DM distribution for number of face-to-face contacts for a typical individual child from Cohort 2. The amalgamated DM distributions are characteristically heavy tailed.

The changes of daily contacts between pre-COVID and new normal (COVID) times can be succinctly summarized by plotting the 5th, mean and 95th values for each cohort or for adult–adult contacts (figure 6). If there had been no change from one regime to the other, then the data would plot along a 1 : 1 line. Lines of the actual percentage change in estimated quantile contact rates for each cohort are shown together with upper and lower bounding changes over all cohorts so that the relative reductions in different cohort contacts can be easily judged.

Responses to specific questions reflected different sources of uncertainty. Some questions concern the contacts of individuals while others concern variations among groups of individuals. Each expert has been asked effectively to make measurements of contacts. The definition of a contact (conversation at 1 m for 5 min or more) is challenging and quite large uncertainty is implicit in making the measurement. Most experts provide large ranges for their estimates of numbers of face-to-face contacts (e.g. figure 3*a* and *b*); these spreads represent intrinsic epistemic uncertainty in making such judgements in the absence of hard empirical data. The post-elicitation interviews conducted with a sub-set of experts confirmed that the teachers were largely thinking about how contacts might vary from day to day, with this being harder to estimate during pre-COVID times due to the diversity of in-school activities pursued by children and adults under pre-pandemic, normal circumstances. In the interviews experts also placed different emphasis on thinking this through; in some instances, 5 min of contact was considered a long time for some children during the course of play-led learning.

In comparing schools and combining experts into a DM, two additional sources of uncertainty arise resulting in the wide DM ranges (figure 4). First there are likely to be real differences between schools because of variations in bubble sizes and school characteristics. As an example gleaned from the interviews, some schools had special-learning units, which meant some children in the cohort spent only part of the day engaged in mainstream learning. In all of the interviews, the experts attested to the strict way in which the bubbles were observed, and thus influenced their thinking around contacts. Second, there is a calibration issue with each expert working out how to make the measurement. These uncertainties are manifest in the wide variation of elicited values observed in

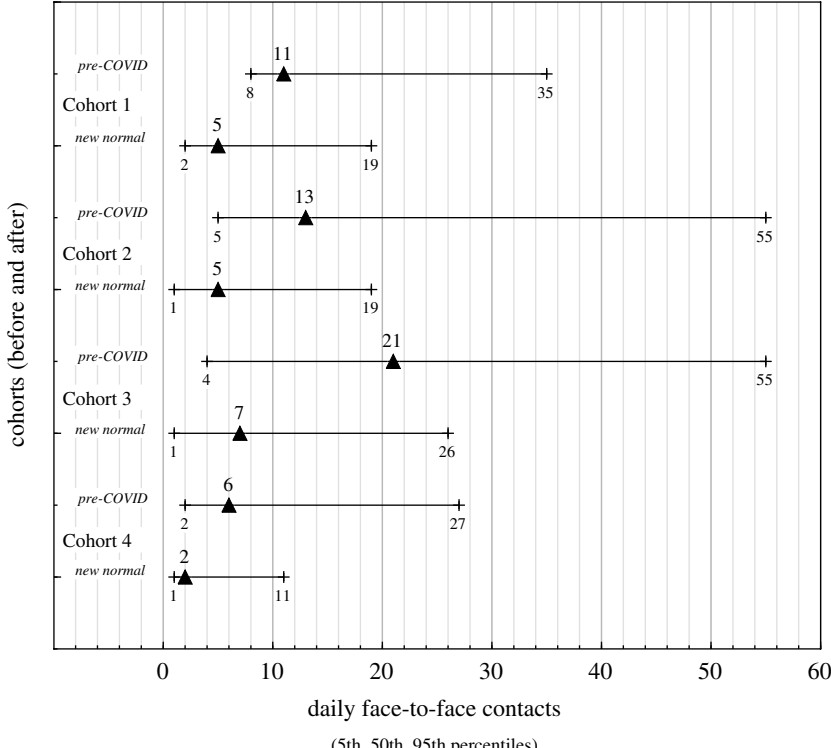

**Figure 4.** Reformulated quantile range graphs for the expected daily number of contacts in each cohort when teachers' median judgements for the 'typical' individual (e.g. figure 3) are combined with their judgements about contact rate variations across different individuals in each cohort, determined from secondary elicitation questions framed in terms of the 'least' and 'most' number of contacts by any individual within a cohort (see text).

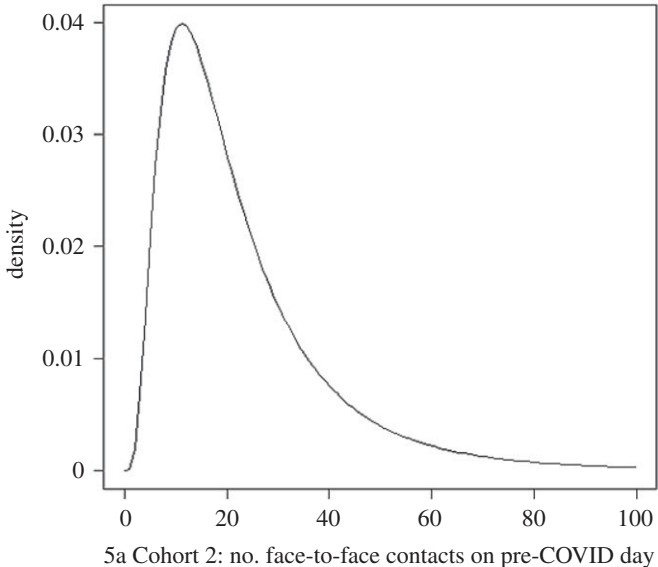

5a Cohort 2: no. face-to-face contacts on pre-COVID day

**Figure 5.** Decision maker (DM) distribution for number of face-to-face contacts of a typical individual child from Cohort 2.

figure 4. In normal circumstances, face-to-face discussions between experts might have been able to reduce the calibration effect. In creating the DM we are combining measurement accuracy at schools, real variations in contacts between schools and significantly different calibrations.

The DM distribution aggregates all these uncertainties and results in markedly skewed distributions (figures 4 and 5). Here, we take the ratio of the right to the left tail, commonly termed eccentricity, as an approximate measure of skewness. Table 6 compares the average eccentricity of the individual experts with the DM eccentricity for several questions. The conflation of all the uncertainties with a DM

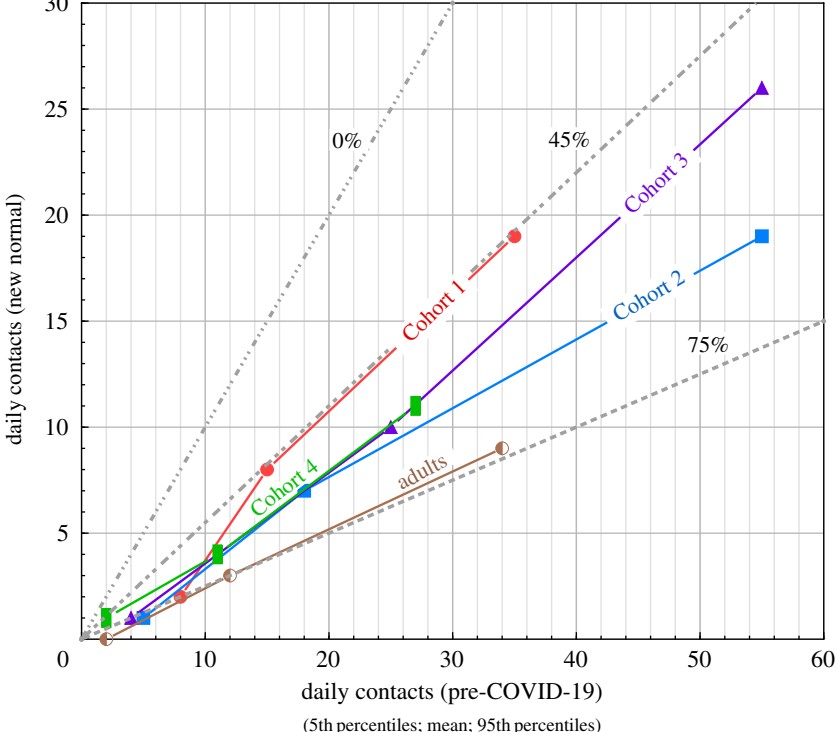

**Figure 6.** Plots of 5th percentile, mean and 95th percentile daily contact numbers for 'pre-COVID' and 'new normal' times from elicited data for the four in-school cohorts and for adult-to-adult contacts in school. A contact is defined as a face-to-face conversation within 1 m for 5 min or more. Individual judgements are synthesized using the item weights algorithm in the classical model. Cohort 1: Nursery, Reception and Year 1 children; Cohort 2: Year 2–6 children (Year 2–5 are vulnerable children and those of key workers); Cohort 3: classroom teachers and teaching assistants; Cohort 4: non-teaching staff (e.g. administrators, cooks, etc., expected to have less frequent contact with children); adults represents within-school adult-to-adult contacts. The dashed lines represent no change in contact rates (0%), and the percentage reductions in daily contacts that bound the elicited data (viz. 45% and 75%).

distribution leads to a quantitative description of the uncertainty of the contacts for a typical or an average pupil, but because it integrates measurement errors, calibration errors and aleatory uncertainty it is not appropriate for developing an epidemiologically relevant model of contacts.

Epidemiological models of transmission convert contact data into a distribution of individual $R_o$ values [19]. Here, we created a DM distribution from data on the median contacts that characterize the average person (Q2a, Q3a, Q5a, Q8a, Q9a, Q10a and Q11a) combined with least and most number of contacts as tails (Q2b, Q3b, Q5b, Q8b, Q9b, Q10b and Q11b) to characterize the distribution of individual contacts within a cohort. We interpret least and most as 1 percentile and 99 percentile values. The interviews confirmed that some teachers worked through the 5 and 95 percentiles by thinking of the most extreme values and then adding even more to account for uncertainty. Thus, each expert's original face-to-face (a) median is coupled with his or her least and most (b) values, with the latter taken as 1 percentile and 99 percentile values. All experts' inputs are processed to form the joint [1; 50; 99] DM for these items. The DM distribution is inspected for its 50th percentile value, and for the corresponding 5th percentile and 95th percentile values, and these are the (c) values reported in table 5 as Q2c, Q3c, Q5c, Q8c, Q9c, Q10c and Q11c, plus their means. We propose that these distributions are the most appropriate for transforming into distributions of individual $R_o$ values for input to epidemiological models.

### 3.2.2. Bubbles

An important risk mitigation strategy [Q1b] is to form groups of children (bubbles) who are organized to learn together to limit contacts with many children. The DfE guidelines issued during the study period were for schools to form bubbles of up to 15 children. The responses indicated bubbles between 6 and 15 with a mean of 11 (Cohort 1) and 13 (Cohort 2). The response to the question on the spacing between

**Table 6.** Ratio of right to left tail (eccentricity) as proxy for skewness contrasting average of individual experts and decision maker.

| item | average expert ratio | DM ratio |
| --- | --- | --- |
| 2a Coh 1 pre-COVID | 1.86 | 9.36 |
| 3a Coh 1 new norm | 1.29 | 4.45 |
| 5a Coh 2 pre-COVID | 1.30 | 3.71 |
| 6a Coh 2 new norm | 1.18 | 3.87 |
| 8a Coh 3 pre-COVID | 1.71 | 5.52 |
| 9a Coh 3 new norm | 1.42 | 8.64 |

bubbles indicated that each bubble was in a separate room and so these results are not reported as they are deemed not relevant to the risk of transmission between children in different bubbles during class time. Expert interviews substantiated this view, with teacher–teacher contacts between bubbles being substantially restricted. The interviewees cited this change of behaviour as one of the most significant in terms of reduction in contacts for teachers.

### 3.2.3. Contact data

We first consider results for contacts of children in normal pre-COVID times (Q2, Q5). Taking the mean values of 25 contacts for Cohort 1 (Q2a) and 18 contacts for Cohort 2 (Q5a) suggests a minimum of 2.1 and 1.5 contact hours. These values are 30–21% of a typical school day (approx. 7 h). The distributions are highly asymmetric with high tail extending to high contact numbers. This feature shows up both in the upper end of the distribution for one child (Q2a and Q5a) and for the values for the most contacts (Q2b and Q5b). The combined central values (Q2c and Q5c) reflect variations among children indicating that a more active child is 2.3 times (Cohort 1) and 3 times (Cohort 2) more active and the least active child 0.5 (Cohort 1) and 0.3 (Cohort 2) less active than a typical (mean) child. The results for Cohorts 1 (Q2) and Cohort 2 (Q5) are similar with a weak indication that younger children have slightly greater contacts compared to older ones and that older children have a somewhat wider range of activity.

Although our approach is fundamentally different to previous contact survey approaches, we find comparable results. Datasets on primary age children are sparse and often out of date. Adults are typically surveyed, although those with children can be included with tailored surveys. The most widely used survey, POLYMOD [3], gives an average of 14.81 (standard deviation of the mean = 10.89) contacts per day for the 5–9 age group ($N = 661$). The Social Contact Survey, conducted in 2010, collected data within schools from 166 children aged 5–9 years. There, a median of 12 contacts per child was measured (mean 14.6, standard deviation 13.7, 95% percentile range 4, 40) [5]. Other studies (e.g. [6]) are based on smaller sample numbers (a few tens) for primary age children and more recent UK surveys do not include people under 18 years of age [21,22]. The data in these studies suggest large variations from the mean and a long tail of high-contact individuals. These studies do not report 5th and 95th percentiles on the contact distributions.

A study using mote devices in four US elementary schools [7] found mean daily contacts (greater than or equal to 5 min) ranging between 25.6 (s.d. 10.3) and 38 (s.d. 13.3). These results are somewhat higher than the SEJ estimates reflecting that motes may record contacts typically up to 3 m away. The teacher-elicited data for pre-COVID times (Q2c and Q5c) are thus qualitatively consistent with previous studies. The sensitivity of contact number data to methodology is, however, highlighted by a study of a French primary school [8], where daily numbers of contacts (of 20 s or more) slightly exceed 300 but only 0.2% if these contacts exceeded 5 min in duration.

The contacts between children in new normal time under risk mitigation regimes are substantially reduced (figure 6): mean contact numbers are reduced by 53% for Cohort 1 (Q3a) and 62% for Cohort 2 (Q6a). The latter reduction is slightly less than the reduction of 74% for adults found by Jarvis et al. [22]. The somewhat greater reduction in contacts in Cohort 2 compared to Cohort 1 in new normal times, however, indicates that older children are slightly easier to manage with respect to physical distancing. This interpretation was supported by interviewees. Some thought that older children were more capable of more sustained (greater than 5 m) physical distancing (talking face-to-face) in free time, enhanced by reduced contacts engendered by the more formal layout of the school environment. Young children

needed more direct contact due to the need for comfort or due to accidents. Long tails indicative of a small number of very active children are still apparent (Q3c and Q6c), but the tails represented by 95th percentile values are reduced by similar amounts to the means. The division into bubbles where class sizes were reduced by factors of 2 and 3 are interpreted to be significant factors in the reduction of contacts, corroborated at interview. This inference is supported too by characterization of contact networks within US elementary, middle and high schools [7], where class size is a major factor in determining daily contacts. Much larger decreases in contacts among children of between 4 and 10 are documented in Zhang *et al*. [6] as a consequence of lockdown. In this case, schools were closed and strict lockdown regimes prevented the mixing of children from different families.

The formation of bubbles, however, is unlikely to be the only factor in the decrease in contacts. Children will naturally form smaller groups in schools due to either juxtaposition in class or the formation of friendship groups that are much smaller than a classroom [23]. Thus, most contacts will be restricted to a fraction of a class in normal times, while in bubbles the size of friendship groups will become more comparable. The interviews indicated, however, that reception classes, in particular, were more free flowing with gregarious children interacting with sibling groups in other years as well as their peer group within bubbles. These observations help explain the significantly smaller reduction in contacts in Cohort 1.

Answers to questions on different cohorts (Q4 and Q7) indicate only weak differences. The data indicate no difference between nursery and reception age children and a weak indication that Year 1 children have slightly fewer contacts. However, the uncertainties are large, indicating low confidence in these judgements. Year 6 are not judged to be different to Years 2–5. Views about this were controlled by the idea that children's behaviour changed gradually and that within any one peer group this varied between children. Nonetheless, there was a fairly uniform view that there was greater potential to mitigate contact behaviour in older children, that changes in class layout had a strong control on 'in class' contacts, and that younger children tended to play side by side but need more close contact with adults. The view that younger children are more difficult to manage than older children is supported by the greater reduction in contacts for Cohort 2.

The responses indicate that classroom staff (Cohort 3) have a mean total contact number of 25 with both children and adults (Q8c). Typically, classes in primary school involve 30 pupils with a single teacher and the same teacher will have ad hoc interactions with other teachers and staff during the day. Converted into the duration of contacts this equates to a minimum of 2.2 contact hours. Ranges of contacts among teachers (Q8c) indicate a large variation. Again, the results indicate some skewness with a heavy tail of large contact numbers. The proportion of children and other adults can be evaluated from Q12 and Q13, noting that the total mean number of contacts from the answers of 32 (Q12 + Q13) can be compared to 25 (Q8c) indicate consistent responses. The results indicate contacts involve 65% children and 35% adults.

Contacts for adults (in the expected age ranges of teachers and teaching assistants) are reported to be about 12–14 per day [3] and approximately 15–20 per day [6]. In the Social Contact Survey, 298 participants were teachers: they reported a median of 18 and mean of 51.2 contacts per day [5]. The mean of 25 daily contacts in our study is consistent with the Social Contact Survey results, taking into account that school hours are only a part of the day and differences in the methods of measuring contact numbers. The high number of contacts within school hours for normal times indicates the interaction with their charges.

In new normal times mean daily contacts for Cohort 3 (Q9c) decrease from 25 to 10, a 60% reduction (figure 6). The reduction in contacts compared to normal times is a factor of 2.5, compared to between 2.1 (Cohort 1) and 1.5 (Cohort 2) for children. Note that the mean of 10 is similar to the bubble size of 11–13 (Q1b). The contacts between adults between normal (Q13) and new normal (Q15) reduces by 80% (figure 6). This result is comparable to a 74% reduction in adults found by Jarvis *et al*. [22]. These results indicate that teachers are physically distancing to a greater extent than children, limiting close contacts as far as is feasible. Answers to Q9b indicate variations in the roles of different teachers and effects of tasks like supervising breaks and meal times. There is still a heavy tail to the responses but this is not as great as for other responses. In the interviews, the experts perceived that teachers facing older children were strongly physically distanced but those facing younger children were less able to strictly observe this.

For Cohort 4 the changes between normal (Q10) and new normal times is similar to Cohort 3, but in general, contacts are fewer (by approx. 30%) and reflect different roles of ancillary staff, some of which involved much less interaction with children (e.g. administrative staff). Responses to Q11 indicate a 64% reduction of contacts in new normal time (figure 6), reflecting the deliberate policy of limiting ancillary staff contacts with children and the efforts of these staff to observe physical distancing. Adult to adult

contacts are reduced by 80% between normal and new normal time (figure 6). At the interview, Cohort 4 were uniformly perceived as adults with the greatest change to their day-to-day contacts in the working day, with some able to be entirely socially distant. This is similar to the overall reduction in contacts of 20% in the general community reported in Brooks-Pollock et al. [24]. Inevitably a school is a workplace where the overall adult to adult contacts are inevitable; the daily contacts within school hours are at the lower end of contact numbers reported in other contact studies [3,5,6].

In figures 4 and 6, the tails of the contact numbers (5th and 95th percentiles) are compared and the overall reductions in contacts are similar to the mean, falling within the 45–0% range. The comparison reduction in contacts due to the risk mitigation arrangements are approximately as effective across the wide range of daily contacts of individual children, classroom staff and support staff.

We asked about adherence and the response indicates that the experts considered adherence to be high (Q16). This view is supported by the analysis of the qualitative parts of the questionnaire related to risk mitigation measures. From the interviews, strong adherence was driven often by thinking about consequences, for children, their families but also their colleagues.

## 3.3. Risk mitigation survey

Questions about risk mitigation measures (table 6) were answered by 23 teachers, except Q1a (21) and Q24 (15). Verbatim individual responses are provided in electronic supplementary material, S1 (§3), including examples of innovation in risk management, mitigation and reduction. Teachers have followed government guidelines (linked to footnotes). However, these have been updated regularly so it is difficult to be sure which guidelines had been followed when teachers completed their elicitations. In instances where guidelines are not defined, teachers have typically been very cautious and adopted measures that suit their own settings.

Answers to Q1a concerning the strategy to reduce close contacts of children were broadly similar for Cohorts 1 and 2. Government guidelines recommend smaller group sizes, with students and teachers kept 2 m apart where possible with staggered break times. Outdoor activities are recommended when possible and students are to stay in the same desks and rooms as much as possible. Seventy-six per cent of responders noted physical distancing measures that were put in place with visual indicators for children to follow. Over 90% of responders indicated classroom bubbles below the DfE recommended 15 (average 10 table 5). In some cases, children were only in school part-time to accommodate reduced class sizes.

Other measures included removal of furniture to free space, rotation or removal of toys and play items, and lunches taking place in the classroom. A third of the responses indicated that learning was moved outdoors as much as possible and a similar number noted the need for individual desks and resources. Around a third of teachers interviewed ensured that bubbles are allocated their own areas of the playground, their own toilets or their own lunchtime spaces and over 50% refer to staggered break times, start times, etc. The interviewees used the word 'strict' to describe their bubbles and indicated that connections between bubbles were discouraged.

SARS-CoV-2 can persist on surfaces for hours to days [25] and disinfecting surfaces is recommended by major health authorities to reduce the chances of infection through touching contaminated surfaces [25]. All responders to Q18 referred to changes to cleaning regimes with over 70% indicating ongoing cleaning throughout the day and highlighting high touch areas such as computers (three responses) and toilets (13 responses), consistent with government guidelines.[3] In the main, deep cleaning took place before and after school following Government guidelines[4] and after classroom bubbles had used specific areas. Three interviewees mentioned sterilization of hard plastic toys following DfE guidelines.[5] Some teachers referred to the removal of soft toys and furnishings in their answer to question Q1a. Three schools employed additional cleaning staff and three reported that staff and children engaged in cleaning. Several interviewees noted that classroom doors were left open, presumably to avoid the need to touch door handles, as recommended in the guidelines.[6] Some responses (electronic supplementary material) indicate measures taken that go beyond the guidelines,

[3]https://www.gov.uk/government/publications/coronavirus-covid-19-implementing-protective-measures-in-education-and-childcare-settings/coronavirus-covid-19-implementing-protective-measures-in-education-and-childcare-settings#effective-infection-protection-and-control.

[4]https://www.gov.uk/government/publications/coronavirus-covid-19-implementing-protective-measures-in-education-and-childcare-settings/coronavirus-covid-19-implementing-protective-measures-in-education-and-childcare-settings#when-open.

[5]https://www.gov.uk/government/publications/preparing-for-the-wider-opening-of-early-years-and-childcare-settings-from-1-june/planning-guide-for-early-years-and-childcare-settings#Section2.

such as: bubbles having their own toilet cubicle and sink; cleaning staff using PPE which are double bagged and stored for 72 h before binning; and disinfecting laptops after every use. Allergies to disinfectant in some children hampered cleaning of some areas.

Answers to Q19 were consistent and included implementing staggered times, one-way systems, different entrances, no adults on site and no playground meeting. Other than different entrances for individual groups, these measures follow recommendations:[7] 65% of responders indicated a staggered drop-off and collection time for children by either year group or class bubbles. Two respondents noted the challenge this is for family groups and had amended their policy accordingly. Thirty-nine per cent had put in place one-way systems to reduce contact between parents and children arriving and those leaving (at drop-off and collection times). Fifty-two per cent noted that parents were either not allowed on the school site or that only one parent could drop the child off, with teachers meeting children at the school gates. Forty-eight per cent reported physical distancing measures for parents and for children in the playground. Approximately 20% referred to children not being allowed in the playground at the start of the school day and having to go straight to class and a further 20% reported classes and bubbles using separate entrances and exits.

Accurate quantification of the risk reduction gain from such measures is difficult. A scoping calculation, however, can give an indicative estimate. If the average parent has daily contact hours of 30 [5] then the mitigation measures described might reduce the contact hours by 1 or 2 h compared to normal mixing associated with delivering children to and from school, so we estimate an overall effect of a few per cent (approx. 3–6%), which can be compared to the 80% reduction associated with general lockdown [24]. The contribution to risk reduction within schools is likely to be very small, but larger and tangible in the wider community.

Responses to the question on weather (Q20) indicate that contact rates were not different between indoors and outdoors. It is now widely thought that outdoors is much less risky than indoors, but this was not the question. Interviews indicated that the physical layout of the school informed their responses, and some commented on the beneficial influence of very good weather.

In relation to hand-washing (Q21), government guidelines recommend that adults and children should frequently wash their hands with soap and water for 20 s and dry thoroughly.[8] Hand-washing reduces transmission of infectious diseases [26–28], but may become counterproductive, less effective and even harmful if hand-washing becomes excessive. Many schools adopted procedures for hand-washing that were age appropriate. However, what is meant by frequently was not specified, leading to considerable variation in interpretation. Responses (table 6) gave a range from 3 to 13 times per day, with over half opting for a range between 3 and 10 hand-washes. The significant reduction has been estimated for between 6 and 10 hand-washes per day [27,28]. Those teachers who gave a specific lower number with a wider range have implemented a policy of hand-washing at certain times with variations added for toilet visits.

The interviewees' schools followed guidelines for pupils or staff displaying COVID19 symptoms (Q22), but some schools have gone beyond these recommendations. In general, anyone showing symptoms is isolated, with some schools having designated areas set aside. A few used PPE, which is recommended if the adult cannot keep 2 m away from pupils. Symptomatic persons are sent home immediately and asked to get tested. School areas are then cleaned, following guidelines[9] to use PPE. Two respondents reported that families of children in a bubble with a symptomatic child would be notified. One school proposed that everyone in a bubble would be recommended to get a test, with the whole bubble being required to self-isolate for 14 days if any test were positive. Another responder stated that a positive test would close the school.

Only a few reported that, for symptoms acquired outside of school, the school should be informed, with a test undertaken and reported. Most interviewees had experienced at least one instance of a suspected case that required testing. While protocols required that the child stayed away, actions

[6]https://www.gov.uk/government/publications/coronavirus-covid-19-implementing-protective-measures-in-education-and-childcare-settings/coronavirus-covid-19-implementing-protective-measures-in-education-and-childcare-settings#when-open.

[7]https://www.gov.uk/government/publications/coronavirus-covid-19-implementing-protective-measures-in-education-and-childcare-settings/coronavirus-covid-19-implementing-protective-measures-in-education-and-childcare-settings#how-to-implement-protective-measures-in-an-education-setting-before-wider-opening-from-1-june.

[8]https://www.gov.uk/government/publications/coronavirus-covid-19-implementing-protective-measures-in-education-and-childcare-settings/coronavirus-covid-19-implementing-protective-measures-in-education-and-childcare-settings.

[9]https://www.gov.uk/government/publications/covid-19-decontamination-in-non-healthcare-settings/covid-19-decontamination-in-non-healthcare-settings.

**Table 7.** Survey of opinions of 18 teachers on how contact patterns will change with a full return to school.

| for children: | |
|---|---|
| I expect the contacts to be the same as new normal times | 1 |
| I expect the contacts to be closer to new normal than normal | 5 |
| I expect the contacts to be half-way between new normal and normal | 8 |
| I expect the contacts to be closer to normal than new normal | 4 |
| I expect the contacts to be the same as normal times | |
| **for adult staff:** | |
| I expect the contacts to be the same as new normal times | |
| I expect the contacts to be closer to new normal than normal | 10 |
| I expect the contacts to be half-way between new normal and normal | 5 |
| I expect the contacts to be closer to normal than new normal | 3 |
| I expect the contacts to be the same as normal times | |

varied in terms of notification or isolation of their bubble until the test result was received. DfE guidance states that the bubbles only close if a positive test is returned but one school required immediate self-isolation until results come back.

Most responders to Q23 (where a parent or relative of child is infected) stated that the child cannot attend school and must self-isolate following government guidelines.[10] One school stated they had no policy for such events and two schools indicated that the child could still come to school.

Other comments (Q24) are unique to individual schools and are all included in the electronic supplementary material. Some noted issues of EAL and SEND students[11] and risk factors involved with school transport.

The structured interviews highlighted some additional issues. Interviewees mentioned the difficulties and stress for staff in maintaining physical distancing and risk mitigation measures. The tension between physical distancing and educational objectives of learning and developing social skills was highlighted. Some commented that measures would be much more difficult with a full return of school. We, therefore, asked the teachers to assess to what extent physical distancing could be maintained with the full return of children in September and the results are given in table 7. While there was a wide range of views, all considered that some physical distancing can be maintained.

# 4. Discussion

The circumstances in English primary schools in June and July 2020 are unprecedented and unlikely to be repeated. The partial re-opening of schools was undertaken under strict guidelines of physical distancing and a range of risk mitigation measures to reduce the transmission of COVID-19. These unique circumstances allowed us to characterize contact patterns for young children and staff in the school environment and the opportunity to evaluate the efficacy of different risk reduction strategies.

We were fortunate that volunteerism led to the participation of knowledgeable people from a group of 34 schools covering a wide range of sizes and communities. Thus, as in opinion polls, an accurate picture of what is happening in schools across England can be gleaned from a modest sample size. Even though the schools individually predicted and then reported a very wide range of returns for pupil and teacher attendance on 1 June, the average for the 34 schools was very close to the picture for England. The prescience of experienced school leaders could be used to anticipate what will happen in September and beyond when a full return to school has been mandated by the Government.

One aspect of our study which warrants discussion is the choice of 5 min as the threshold for a significant contact. Our study misses out on shorter more frequent contacts. The study of US

---

[10]https://www.gov.uk/government/publications/covid-19-stay-at-home-guidance/guidance-for-households-with-grandparents-parents-and-children-living-together-where-someone-is-at-increased-risk-or-has-symptoms-of-coronavirus-cov.

[11]EAL are students with English as a second language and SEND are those with special educational needs and disabilities.

elementary schools [7] found that short frequent contacts made up, averaged over four study schools, 64% (less than or equal to 5 min) and 38% of contacts (less than or equal to 1 min). Such short contacts will contribute to the total daily contact duration. We can illustrate the contribution of short frequent contacts to the total duration by considering the median of 25 daily contacts for Cohort 1 in normal times. Assuming the contact duration distributions of [7] are representative, then approximately 1/3 of contacts are greater than or equal to 5 min, 1/3 are between 1 and 5 min and 1/3 are 1 min or less. Noting the power-law character of contact durations [7] a logarithmic mean of 0.3 min (less than or equal to 1 min) and 3 min (greater than 1 to less than 5 min) for the short frequent contacts leads to approximate durations of 7.5 min of 1 min and 75 min for contacts between 1 and 5 min compared to greater than 125 min for contacts during one school day. We conclude that contacts greater than or equal to 5 min contribute the majority of the total duration of close contacts.

Breaking social networks is a key non-pharmaceutical intervention for countering the spread of infectious disease. Our study indicates that contacts within schools were reduced between 45% and 80% (figure 6). These strikingly successful outcomes highlight the tremendous work of school staff and the role of greatly reduced class sizes through the creation of bubbles of children which are much less than normal class sizes. Although the marked reduction in contacts can be partly explained by much smaller class sizes, observations indicate that social groups are characteristically smaller than typical class sizes under normal times [24]. Thus, the organization of children within bubbles of groups combined with control of their behaviour by teaching staff is a significant factor. A combination of 3/7 of children being invited back to school and limited parental compliance returning eligible children to school on 1 June (30–40%) led to typically only 15–20% of children being present. This allowed schools to form bubble sizes significantly less the 15 indicated by DfE guidelines (range 7–16 with median of 11). Contact numbers do not scale linearly with group size and the number of contacts saturates at about 20 [24]. The bubble sizes of about 10 have proved to be efficacious in decreasing contacts, limiting mixing between children and thus reducing risk of transmission.

Contacts between adults and children have been reduced by a factor of about 5× between normal and new normal times. The contacts of teaching staff (Cohort 3) decreases by 50% between normal and new normal times indicating that adults can more effectively adopt stringent social distancing practices. A reduction in contacts of about 80% is also approximately the average achieved in the UK at the height of the lockdown [18]. Given that mixing with groups of children is part of the job of classroom staff, the reduction is impressive. Likewise, other adult staff achieved similar reductions in contacts. The data also confirm that older children (Cohort 2) are somewhat easier to manage than younger children (Cohort 1).

Notwithstanding distinct and contrasting methodologies, the elicited data accord well with other kinds of contact survey data. The results indicate similar heterogeneity as documented in Danon *et al*. [5] in which the data are a mixture of individual close contacts plus contacts related to groups where each person in a group counts as a contact. Results were expressed as contacts per day and total contact hours, noting that the latter parameter can exceed 24 h because contacts within groups occur simultaneously. In a school setting, classroom adults (Cohort 3) have a median of 26 contacts (Q8a), of which about 2/3 are with children (Q12) and 1/3 with adults (Q13). The distance specified in the question is 1 m with physical distancing at the time of elicitation being 2 m. Contact hours are of the order of 2 h. In the context of a typical class of 30, the group definition of contacts in Danon *et al*. [5] seems less relevant. There must be an additional risk factor related to being in an enclosed space with widespread aerosol circulation, but this is not quantifiable with the elicitation data. However, reducing class sizes from 30 to roughly 10 reduces the random risk of an infectious person being in the classroom by about a third.

We recognize that in a school setting close contacts of individuals may not be the only factor in controlling transmission. In particular, very fine particles with long atmospheric residence times may play a contributory role in airborne transmission in enclosed spaces with poor ventilation [29–31]. At least qualitatively, this risk is expected to be proportional to the number of people in a room and the duration they remain there. Since contact statistics are also expected to be closely related to these same factors, forming bubbles with fewer persons in a classroom will reduce risk for both mechanisms of transmission.

All of the risk mitigation strategies contribute to the reduction of risk. Specific risk management strategies reflect individual circumstances in schools. Many schools went beyond official DfE guidelines. Many relate to steps which are likely to reduce contacts and disrupt break-time contact networks. Each risk mitigation measure will contribute to risk reduction, but it is hard to quantify precise impacts.

It seems unlikely that the significant reduction of risk, implied by these results, can be maintained with a full return to school without greatly expanding the accommodation to maintain reduced class

sizes, as suggested by the factor of 2–3 reduction in contacts between children. Adult staff can continue to observe strict physical distancing behaviours and can continue to organize the classroom and break times which reduces contacts. While opinions vary widely (table 7), there is a broad consensus that physical distancing measures can be maintained to some extent with a full return of children, but not to the same extent as achieved in June and July.

In summary, we have applied an alternative methodology to generate initial estimates of in-school contact rates for children and adults. Our findings contribute to filling significant epistemic knowledge gaps for primary schools in England, while also adding quantitative information about the intrinsic aleatory variability of such contact rates. These new contact data are similar to the limited data derived using other methods, such as surveys and use of electronic motes. We have shown through SEJ that daily contacts were significantly reduced in English primary schools during a partial return to school. Our results provide the basis to establish variations of localized $R_o$ values associated with individual schools for input into epidemiological and probabilistic transmission models. Our study also documents the effectiveness of mitigation strategies in schools that complement social distancing policies.

Ethics. All teachers involved in the elicitation were volunteers recruited through the Royal Society Schools Network. The identity of the schools and teachers (experts) has not been disclosed and all responses have been anonymized. The protocols for the interviews adhered to the University of East Anglia ethics procedures.

Data accessibility. All data are contained within the electronic supplementary material.

Authors' contributions. R.S.J.S. and W.P.A. designed and coordinated the project and led the writing of the paper. R.M.C. supported the design of the elicitation and interpretation of elicited data. E.B.-P. and L.D. provided input into the elicitation questions on contacts and interpretations of the contact data. J.B. carried out the structured interviews of experts. J.H.S. supported the project through fact finding. J.C. coordinated the school and teacher input and led on analysis of the responses to the risk mitigation survey. All authors contributed to the interpretation of the results.

Competing interests. We declare we have no competing interests.

Funding. The support of the Royal Society education policy unit and the RAMP initiative for COVID-19 are acknowledged. Funding to L.D. and E.B.-P. from MRC grants MR/V038613/1 and MC-PC-19067 is acknowledged.

Acknowledgements. We much appreciate the primary school leaders who volunteered to join the expert panel. Their enthusiasm, support and expert knowledge were paramount. The University of East Anglia fact-finding team of Jade Eyles, James Christie, Nicola Taylor and Martin Mangler are much appreciated. The project was completed under the auspices of the RAMP initiative of the Royal Society. Mike Cates is thanked for encouragement and advice. Two anonymous and constructive reviews led to significant improvements in the paper.

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
