## [Reviewer comments · Royal Society Open Science]

Review History

RSOS-201566.R0 (Original submission)

Review form: Reviewer 1

Is the manuscript scientifically sound in its present form?

Yes

Are the interpretations and conclusions justified by the results?

Yes

Is the language acceptable?

Yes

Do you have any ethical concerns with this paper?

No

Have you any concerns about statistical analyses in this paper?

No

Recommendation?

Accept with minor revision (please list in comments)

Comments to the Author(s)

In this study, Sparks et al. conducted a structured expert elicitation to address an important gap in data on current contact rates and patterns for students and staff in primary schools in the UK. Their work helps to provide modern estimates of the mean number of contacts between students and staff stratified into different cohorts and contacts between different individuals in different groups (younger students, older primary school students, teaching staff, non teaching staff). Sparks et al. correctly identify that some of the last data collected on this aspect of contact patterns is from well over a decade ago and more recent data should be collected to assess the risk of transmission of COVID-19 through contact in schools. Their work also helps to identify the real world reduction of contact from current risk mitigation strategies in place in UK schools. Collection and publication of this type of data is highly valuable not only during pandemic times but also for future reference to help us better plan in the event of future outbreaks.

Before publication I think this article could benefit from revisions with my comments as follows.

First, the article refers to schools across the UK, however 2 volunteer experts were dropped due to being in Wales or Scotland. This would seem to mean that the experts don't represent the whole of the UK, but rather only England and possibly Northern Ireland. The language of the article should change to be more specific. It does not seem like Northern Ireland was represented among the experts either, so if this is the case I would suggest the authors instead say that the study represents schools in England only.

Second, again the article refers to contact patterns in the UK and the general lack of data on contact patterns of younger children in the school setting. I can understand only focusing on studies in the UK, however, there is a 2016 study (Guclu et al.) on contact patterns of children in schools in the US that does include younger children to a great extent. I am surprised to not see this article referenced or compared to at all. While the countries are different, school structures and mixing patterns in developed Western countries do share many similarities and would be appropriate to make some comparisons. I would suggest the authors look at this work and see how their expert judgements of pre-COVID contact rates and patterns compare with this study to help understand how their work is applicable beyond the UK/England context.

Third, the authors mention that "(T)he Classical Model for SEJ has been deployed in several public health policy applications [9-15] but is not widely used in epidemiology". Can you explain a bit more on why this approach hasn't been widely used in the field? Is there a legitimate concern among epidemiologists in using this approach? Is this a result of lack of familiarity with this approach in epidemiology?

It's briefly mentioned that volunteer experts are from the Royal Society Schools network and have STEM backgrounds. I think it would be useful to include a statement on how this might bias the results of the expert opinions. Are these educators and staff from schools considered better than average? Do these schools have fewer children of key workers and/or from vulnerable environments? Do Royal Society Schools network volunteers skew towards higher performing schools or do they come from a broad spectrum of schools in the UK?

I found lines 120-123 a bit confusing. Clarification here would be great. Specifically are you saying that 28 experts completed the calibration questions and 26 of those also answered the 2nd questionnaire? It's not clear to me what was meant here.

The authors also state that the schools of the experts surveyed are nationally representative because the pupil to teacher ratios are in line with national averages. However, they also state that the sizes of the schools of these experts are 30% greater than the national average. I don't think you can say that the schools of the experts surveyed are nationally representative then in general. I would simply ask the language change to be more specific.

Line 176 has a period in red text.

In the Methods section there is a value referred to as a 'p-value'. However, this isn't a p-value in the usual sense and I think it would be better to simply refer to it by another name to avoid possible confusion (this is mentioned in the text but changing the name would probably be better).

Overall, if the issues raised can be addressed I think the article can be recommended for publication. Some of the material could find its way instead to the supplementary material if anything. My last recommendation would be that the discussion ends on a stronger note. In its current form it ends in a limitation rather than driving home the message of the article.

Review form: Reviewer 2

Is the manuscript scientifically sound in its present form?

Yes

Are the interpretations and conclusions justified by the results?

Yes

Is the language acceptable?

No

Do you have any ethical concerns with this paper?

No

Have you any concerns about statistical analyses in this paper?

No

Recommendation?

Accept with minor revision (please list in comments)

Comments to the Author(s)

I must preface my review by saying that I am a close colleague and personal friend of one of the authors (Cooke), know one of the others (Aspinall), and am involved in an active proposal with them for research funding from the US National Science Foundation.

This paper uses a novel approach to address a problem of great practical significance -- i.e., the efficacy of mitigation approaches intended to reduce contact in primary schools in England during the pandemic.

The method -- formally-elicited structured expert judgment -- is deftly applied and properly interpreted by some of its leading practitioners.

The paper is, in general, quite clearly written. However some sections (e.g., Results -- especially section 3.2 Elicitation Data) are not. In my view, effort to carefully rewrite these sections would increase the impact of the work.

The extent to which contact, as opposed to airborne transmission, is important in controlling the spread of COVID-19, is an open question. But, this paper contributes important information about the extent of contact in schools and the efficacy of mitigation measures in reducing contact.

In my view the paper would be much stronger if the authors focused on storytelling -- what questions does the paper address? what methods does it rely on to answer these? ... and, perhaps most importantly, what are the answers?

As now written, the manuscript seems to get distracted in some sections on technical details -- which are often not clearly explained.

I believe that an effort to rewrite the paper with this in mind would pay off in terms of broad accessibility and ultimate impact.

I wonder whether in the description of the elicitation results it might not be better to carefully explain the results for one cohort (for example, discussing in some detail the interpretation of responses to questions 2a, 2b and 2c) ... and then to relegate the presentation of the detailed responses to similar questions for the other cohorts to a technical appendix.

One technical concern that I have is that uncertainty and variability are not clearly distinguished.

I know that the authors are well aware of the issue. I would urge them to revise the paper to make clear whether the distributions produced in question 2c, 3c, 5c, 6c, 8c, 9c, 10c and 11c -- reflect variability of of contact or a mix of variability of and uncertainty about contact.

It was also unclear to me whether the variability is variability across students, days or student-days.

----- NOTE -- Detailed Editorial Comments Below -----

Examples of places where the writing could be improved include:

- line 95 ... would be nice to know how many, rather than 'most'
- lines 121-123 ... possibly a word or two missing from this sentence
- line 154 and Table 1 ... in the elicitation of expert judgment a key issue is whether the elicitation questions satisfy the 'clairvoyance' criterion -- it is not completely clear to me that all of

the questions satisfy clairvoyance -- e.g., question 2b -- 'give a range of contacts' -- I'm sure that, as leading experts in the field, the authors have thought carefully about this

-- lines 257-260 and Fig 2 ... not clear why experts were shown in this particular order on the figure ... possibly better if ordered by performance score

-- first few para of section 3.2 are quite dense and opaque

-- lines 284-286 ... since these questions are at the heart of the study ... this needs to be explained much more clearly

-- lines 286-287 and Fig 3 ... one sentence 'shows examples of responses to illustrate variation in responses among experts' -- seems like an inadequate explanation of six highly detailed graphs -- either drop 5 of the 6 graphs or do a better job of explaining these

--lines 287-288 and Fig 4 -- tremendous figure ...key result not clear why the word 'Cases' was used to describe the different cohorts ...also not clear that the 5th, 50th and 95th here refer to variability, rather than uncertainty ... would be worth clarifying.

--lines 289-291 and Fig 6 -- important figure ... but very busy and quite poorly explained ... almost uninterpretable ... what are the 0, 45 and 75% lines? ... do the 5%, mean, and 95% listed parenthetically on the x-axis label reflect variability in the contact rates within the various cohorts or uncertainty or both?/ ... this would seem to be one of the key results of the study and deserves more careful explanation.

-- lines 292-295 ... the meaning of 'The interviews conducted in a smaller sample of experts' is unclear

-- lines 329-342 ... the methodology here is central to the construction of distribution of individual R_0 values ... but this para leaves some ambiguity about exactly what was done ... since the tails of these contact distributions may be important, the details here matter.

-- lines 356-367 ... make clear whether the unit of observation is the student, the day or the student-day ... if this does not matter for the intended use then make this clear as well

-- lines 368-381 ... were the people you interviewed aware of this literature? did you ask the subjects to explain the evidence they relied on and why they chose to emphasize certain evidence and give less emphasis to other evidence ... some would argue that understanding the reasoning of the experts is as important as the quantitative results of the elicitation.

-- lines 598-606 -- it is great that the schools you studied seem in some ways -- 'size and community' -- to be similar in these ways to the broad population of elementary schools in England -- but it is not obvious that they are similar in terms of socioeconomic status of pupils, parents or staff -- all of which could influence the resources available to respond to and the behaviours necessary to adapt to the pandemic.

-- lines 633-640 -- seem not entirely clear to me

Decision letter (RSOS-201566.R0)

Dear Dr Sparks

The Editors assigned to your paper RSOS-201566 "Novel approach to evaluate contact patterns and risk mitigation for COVID-19 in English Primary Schools: application of Structured Expert Judgement" have now received comments from reviewers and would like you to revise the paper in accordance with the reviewer comments and any comments from the Editors. Please note this decision does not guarantee eventual acceptance.

Please submit your revised manuscript and required files (see below) no later than 21 days from today's (ie 13-Oct-2020) date. Note: the ScholarOne system will 'lock' if submission of the revision is attempted 21 or more days after the deadline. If you do not think you will be able to meet this deadline please contact the editorial office immediately.

on behalf of Kevin Padian (Subject Editor)
openscience@royalsociety.org

Associate Editor Comments to Author:

Comments to the Author:

Thank you for the somewhat unusual but interesting submission. The referees are broadly in favour of acceptance once a number of changes have been made, but given the nature of the changes we would like you to have 3 weeks to revise the paper. Note that the revision will be returned to the referees for further comment - they must be satisfied that your revision is ready for publication, and further revisions are unlikely.

Reviewer comments to Author:

Reviewer: 1

Comments to the Author(s)

In this study, Sparks et al. conducted a structured expert elicitation to address an important gap in data on current contact rates and patterns for students and staff in primary schools in the UK. Their work helps to provide modern estimates of the mean number of contacts between students and staff stratified into different cohorts and contacts between different individuals in different groups (younger students, older primary school students, teaching staff, non teaching staff).

Sparks et al. correctly identify that some of the last data collected on this aspect of contact patterns is from well over a decade ago and more recent data should be collected to assess the risk of transmission of COVID-19 through contact in schools. Their work also helps to identify the real world reduction of contact from current risk mitigation strategies in place in UK schools. Collection and publication of this type of data is highly valuable not only during pandemic times but also for future reference to help us better plan in the event of future outbreaks.

Before publication I think this article could benefit from revisions with my comments as follows.

First, the article refers to schools across the UK, however 2 volunteer experts were dropped due to being in Wales or Scotland. This would seem to mean that the experts don't represent the whole of the UK, but rather only England and possibly Northern Ireland. The language of the article should change to be more specific. It does not seem like Northern Ireland was represented among the experts either, so if this is the case I would suggest the authors instead say that the study represents schools in England only.

Second, again the article refers to contact patterns in the UK and the general lack of data on contact patterns of younger children in the school setting. I can understand only focusing on studies in the UK, however, there is a 2016 study (Guclu et al.) on contact patterns of children in schools in the US that does include younger children to a great extent. I am surprised to not see this article referenced or compared to at all. While the countries are different, school structures and mixing patterns in developed Western countries do share many similarities and would be appropriate to make some comparisons. I would suggest the authors look at this work and see how their expert judgements of pre-COVID contact rates and patterns compare with this study to help understand how their work is applicable beyond the UK/England context.

Third, the authors mention that "(T)he Classical Model for SEJ has been deployed in several public health policy applications [9-15] but is not widely used in epidemiology". Can you explain a bit more on why this approach hasn't been widely used in the field? Is there a legitimate concern among epidemiologists in using this approach? Is this a result of lack of familiarity with this approach in epidemiology?

It's briefly mentioned that volunteer experts are from the Royal Society Schools network and have STEM backgrounds. I think it would be useful to include a statement on how this might bias the results of the expert opinions. Are these educators and staff from schools considered better than average? Do these schools have fewer children of key workers and/or from vulnerable environments? Do Royal Society Schools network volunteers skew towards higher performing schools or do they come from a broad spectrum of schools in the UK?

I found lines 120-123 a bit confusing. Clarification here would be great. Specifically are you saying that 28 experts completed the calibration questions and 26 of those also answered the 2nd questionnaire? It's not clear to me what was meant here.

The authors also state that the schools of the experts surveyed are nationally representative because the pupil to teacher ratios are in line with national averages. However, they also state that the sizes of the schools of these experts are 30% greater than the national average. I don't think you can say that the schools of the experts surveyed are nationally representative then in general. I would simply ask the language change to be more specific.

Line 176 has a period in red text.

In the Methods section there is a value referred to as a 'p-value'. However, this isn't a p-value in the usual sense and I think it would be better to simply refer to it by another name to avoid

possible confusion (this is mentioned in the text but changing the name would probably be better).

Overall, if the issues raised can be addressed I think the article can be recommended for publication. Some of the material could find its way instead to the supplementary material if anything. My last recommendation would be that the discussion ends on a stronger note. In its current form it ends in a limitation rather than driving home the message of the article.

Reviewer: 2

Comments to the Author(s)

I must preface my review by saying that I am a close colleague and personal friend of one of the authors (Cooke), know one of the others (Aspinall), and am involved in an active proposal with them for research funding from the US National Science Foundation.

This paper uses a novel approach to address a problem of great practical significance -- i.e., the efficacy of mitigation approaches intended to reduce contact in primary schools in England during the pandemic.

The method -- formally-elicited structured expert judgment -- is deftly applied and properly interpreted by some of its leading practitioners.

The paper is, in general, quite clearly written. However some sections (e.g., Results -- especially section 3.2 Elicitation Data) are not. In my view, effort to carefully rewrite these sections would increase the impact of the work.

The extent to which contact, as opposed to airborne transmission, is important in controlling the spread of COVID-19, is an open question. But, this paper contributes important information about the extent of contact in schools and the efficacy of mitigation measures in reducing contact.

In my view the paper would be much stronger if the authors focused on storytelling -- what questions does the paper address? what methods does it rely on to answer these? ... and, perhaps most importantly, what are the answers?

As now written, the manuscript seems to get distracted in some sections on technical details -- which are often not clearly explained.

I believe that an effort to rewrite the paper with this in mind would pay off in terms of broad accessibility and ultimate impact.

I wonder whether in the description of the elicitation results it might not be better to carefully explain the results for one cohort (for example, discussing in some detail the interpretation of responses to questions 2a, 2b and 2c) ... and then to relegate the presentation of the detailed responses to similar questions for the other cohorts to a technical appendix.

One technical concern that I have is that uncertainty and variability are not clearly distinguished.

I know that the authors are well aware of the issue. I would urge them to revise the paper to make clear whether the distributions produced in question 2c, 3c, 5c, 6c, 8c, 9c, 10c and 11c -- reflect variability of of contact or a mix of variability of and uncertainty about contact.

It was also unclear to me whether the variability is variability across students, days or student-days.

----- NOTE -- Detailed Editorial Comments Below -----

Examples of places where the writing could be improved include:

- line 95 ... would be nice to know how many, rather than 'most'
- lines 121-123 ... possibly a word or two missing from this sentence
- line 154 and Table 1 ... in the elicitation of expert judgment a key issue is whether the elicitation questions satisfy the 'clairvoyance' criterion -- it is not completely clear to me that all of the questions satisfy clairvoyance -- e.g., question 2b -- 'give a range of contacts' -- I'm sure that, as leading experts in the field, the authors have thought carefully about this
- lines 257-260 and Fig 2 ... not clear why experts were shown in this particular order on the figure ... possibly better if ordered by performance score
- first few para of section 3.2 are quite dense and opaque
- lines 284-286 ... since these questions are at the heart of the study ... this needs to be explained much more clearly
- lines 286-287 and Fig 3 ... one sentence 'shows examples of responses to illustrate variation in responses among experts' -- seems like an inadequate explanation of six highly detailed graphs -- either drop 5 of the 6 graphs or do a better job of explaining these
- lines 287-288 and Fig 4 -- tremendous figure ...key result not clear why the word 'Cases' was used to describe the different cohorts ...also not clear that the 5th, 50th and 95th here refer to variability, rather than uncertainty ... would be worth clarifying.
- lines 289-291 and Fig 6 -- important figure ... but very busy and quite poorly explained ... almost uninterpretable ... what are the 0, 45 an 75% lines? ... do the 5%, mean, and 95% listed parenthetically on the x-axis label reflect variability in the contact rates within the various cohorts or uncertainty or both?/ ... this would seem to be one of the key results of the study and deserves more careful explanation.
- lines 292-295 ... the meaning of 'The interviews conducted in a smaller sample of experts' is unclear
- lines 329-342 ... the methodology here is central to the construction of distribution of individual R_0 values ... but this para leaves some ambiguity about exactly what was done ... since the tails of these contact distributions may be important, the details here matter.
- lines 356-367 ... make clear whether the unit of observation is the student, the day or the student-day ... if this does not matter for the intended use then make this clear as well
- lines 368-381 ... were the people you interviewed aware of this literature? did you ask the subjects to explain the evidence they relied on and why they chose to emphasize certain evidence and give less emphasis to other evidence ... some would argue that understanding the reasoning of the experts is as important as the quantitative results of the elicitation.
- lines 598-606 -- it is great that the schools you studied seem in some ways -- 'size and community' -- to be similar in these ways to the broad population of elementary schools in England -- but it is not obvious that they are similar in terms of socioeconomic status of pupils, parents or staff -- all of which could influence the resources available to respond to and the behaviours necessary to adapt to the pandemic.
- lines 633-640 -- seem not entirely clear to me

===PREPARING YOUR MANUSCRIPT===

- one version identifying all the changes that have been made (for instance, in coloured highlight, in bold text, or tracked changes);
- a 'clean' version of the new manuscript that incorporates the changes made, but does not highlight them.

This version will be used for typesetting if your manuscript is accepted.

===PREPARING YOUR REVISION IN SCHOLARONE===

-- If you have uploaded ESM files, please ensure you follow the guidance at <https://royalsociety.org/journals/authors/author-guidelines/#supplementary-material> to include a suitable title and informative caption. An example of appropriate titling and captioning may be found at https://figshare.com/articles/Table_S2_from_Is_there_a_trade-off_between_peak_performance_and_performance_breadth_across_temperatures_for_aerobic_sc_ope_in_teleost_fishes_/3843624.

Author's Response to Decision Letter for (RSOS-201566.R0)

See Appendix A.

RSOS-201566.R1 (Revision)

Review form: Reviewer 2

Is the manuscript scientifically sound in its present form?

Yes

Are the interpretations and conclusions justified by the results?

Yes

Is the language acceptable?

Yes

Do you have any ethical concerns with this paper?

No

Have you any concerns about statistical analyses in this paper?

No

Recommendation?

Accept as is

Comments to the Author(s)

The authors have done an outstanding job responding to review comments. The paper is now excellent and, in my view, should be published without further delay.

The only point that I would like them to make a bit more clearly in the discussion is the possible role of airborne transmission through very fine particles with long atmospheric residence times. At least here in the US, the importance of reviewing and, in many cases, dramatically improving air filtration in school ventilation systems is receiving increased emphasis.

Decision letter (RSOS-201566.R1)

Dear Dr Sparks

On behalf of the Editors, we are pleased to inform you that your Manuscript RSOS-201566.R1 "Novel approach to evaluate contact patterns and risk mitigation for COVID-19 in English Primary Schools: application of Structured Expert Judgement" has been accepted for publication in Royal Society Open Science subject to minor revision in accordance with the referees' reports. Please find the referees' comments along with any feedback from the Editors below my signature.

Please submit your revised manuscript and required files (see below) no later than 7 days from today's (ie 06-Jan-2021) date. Note: the ScholarOne system will 'lock' if submission of the revision is attempted 7 or more days after the deadline. If you do not think you will be able to meet this deadline please contact the editorial office immediately.

Please note article processing charges apply to papers accepted for publication in Royal Society Open Science (<https://royalsocietypublishing.org/rsos/charges>). Charges will also apply to papers transferred to the journal from other Royal Society Publishing journals, as well as papers submitted as part of our collaboration with the Royal Society of Chemistry

(<https://royalsocietypublishing.org/rsos/chemistry>). Fee waivers are available but must be requested when you submit your revision (<https://royalsocietypublishing.org/rsos/waivers>).

on behalf of Prof Kevin Padian (Subject Editor)
openscience@royalsociety.org

Associate Editor Comments to Author:

Thank you for the patience while we reviewed the revision - unfortunately, we had hoped the second reviewer from the earlier round of review would be available to assess the manuscript, but they have not been able to assist. Regrettably, confirming this took longer than we (and no doubt you) would have preferred - please accept our apologies for this.

In any case, we're pleased that the reviewer who has commented on the work has recommended the paper be accepted, albeit while suggesting a few minor tweaks - given they have made some further comments, we'd like you to consider making the changes they recommend and incorporating them into a final revision.

Reviewer comments to Author:
Reviewer: 2

Comments to the Author(s)

The authors have done an outstanding job responding to review comments. The paper is now excellent and, in my view, should be published without further delay.

The only point that I would like them to make a bit more clearly in the discussion is the possible role of airborne transmission through very fine particles with long atmospheric residence times. At least here in the US, the importance of reviewing and, in many cases, dramatically improving air filtration in school ventilation systems is receiving increased emphasis.

===PREPARING YOUR MANUSCRIPT===

Your revised paper should include the changes requested by the referees and Editors of your manuscript. You should provide two versions of this manuscript and both versions must be provided in an editable format:
one version identifying all the changes that have been made (for instance, in coloured highlight, in bold text, or tracked changes);
a 'clean' version of the new manuscript that incorporates the changes made, but does not highlight them. This version will be used for typesetting.
Please ensure that any equations included in the paper are editable text and not embedded images.

Please ensure that you include an acknowledgements' section before your reference list/bibliography. This should acknowledge anyone who assisted with your work, but does not

qualify as an author per the guidelines at <https://royalsociety.org/journals/ethics-policies/openness/>.

===PREPARING YOUR REVISION IN SCHOLARONE===

- Ensure that your data access statement meets the requirements at <https://royalsociety.org/journals/authors/author-guidelines/#data>. You should ensure that you cite the dataset in your reference list. If you have deposited data etc in the Dryad repository, please only include the 'For publication' link at this stage. You should remove the 'For review' link.
- If you are requesting an article processing charge waiver, you must select the relevant waiver option (if requesting a discretionary waiver, the form should have been uploaded at Step 3 'File upload' above).
- If you have uploaded ESM files, please ensure you follow the guidance at <https://royalsociety.org/journals/authors/author-guidelines/#supplementary-material> to include a suitable title and informative caption. An example of appropriate titling and captioning may be found at https://figshare.com/articles/Table_S2_from_Is_there_a_trade-off_between_peak_performance_and_performance_breadth_across_temperatures_for_aerobic_scope_in_teleost_fishes_/3843624.

Author's Response to Decision Letter for (RSOS-201566.R1)

See Appendix B.

Decision letter (RSOS-201566.R2)

Dear Dr Sparks,

It is a pleasure to accept your manuscript entitled "Novel approach to evaluate contact patterns and risk mitigation for COVID-19 in English Primary Schools: application of Structured Expert Judgement" in its current form for publication in Royal Society Open Science.

COVID-19 rapid publication process:

We are taking steps to expedite the publication of research relevant to the pandemic. If you wish, you can opt to have your paper published as soon as it is ready, rather than waiting for it to be published the scheduled Wednesday.

This means your paper will not be included in the weekly media round-up which the Society sends to journalists ahead of publication. However, it will still appear in the COVID-19 Publishing Collection which journalists will be directed to each week (<https://royalsocietypublishing.org/topic/special-collections/novel-coronavirus-outbreak>).

If you wish to have your paper considered for immediate publication, or to discuss further, please notify openscience_proofs@royalsociety.org and press@royalsociety.org when you respond to this email.

You can expect to receive a proof of your article in the near future. Please contact the editorial office (openscience@royalsociety.org) and the production office (openscience_proofs@royalsociety.org) to let us know if you are likely to be away from e-mail contact – if you are going to be away, please nominate a co-author (if available) to manage the proofing process, and ensure they are copied into your email to the journal.

Best regards,

on behalf of Professor Kevin Padian (Subject Editor)
openscience@royalsociety.org

Appendix A

Dear Editor,

On behalf of my co-authors I am submitting a revised version of the paper entitled “A novel approach for evaluating contact patterns and risk mitigation strategies for COVID-19 in English Primary Schools with application of Structured Expert Judgement”. First of all we were pleased to receive two very positive reviews with many helpful suggestions for improvement. We have gone systematically through all comments and provide a document with responses to each point. We include the original text with track changes to make it clear where revisions have been made and a clean version with all revisions accepted.

The main changes have involved clarifications and including reference to and comparison with the important paper by Guclu et al suggested by reviewer 1. Some diagrams have been modified following the reviewers’ suggestions. The most significant change is some re-organisation and expansion of the sections describing the elicitation questions and the results. Reviewer 2 was justifiably critical that the results sections was rather cryptic and dense. We came to realise that we had not explained the context for the elicitation questions or described and justified their composition in sufficient detail, making the section on the results difficult to follow. We have thus largely rewritten section 2.3.1 and have reorganized parts of section 3 on results. Together with a little more explanatory detail on some of the Figures, we hope we have fully addressed Reviewer 2’s comments. Please find below the reviews and detailed responses to each point.

Yours sincerely,

Steve Sparks

Reviewer comments to Author:

Reviewer: 1

Comments to the Author(s)

In this study, Sparks et al. conducted a structured expert elicitation to address an important gap in data on current contact rates and patterns for students and staff in primary schools in the UK. Their work helps to provide modern estimates of the mean number of contacts between students and staff stratified into different cohorts and contacts between different individuals in different groups (younger students, older primary school students, teaching staff, non teaching staff). Sparks et al. correctly identify that some of the last data collected on this aspect of contact patterns is from well over a decade ago and more recent data should be collected to assess the risk of transmission of COVID-19 through contact in schools. Their work also helps to identify the real world reduction of contact from current risk mitigation strategies in place in UK schools. Collection and publication of this type of data is highly valuable not only during pandemic times but also for future reference to help us better plan in the event of future outbreaks.

We thank the reviewer for these positive remarks

Before publication I think this article could benefit from revisions with my comments as follows.

First, the article refers to schools across the UK, however 2 volunteer experts were dropped due to being in Wales or Scotland. This would seem to mean that the experts don’t represent the whole of the UK, but rather only England and possibly Northern Ireland. The language of the article should change to be more specific. It does not seem like Northern Ireland was represented among the experts either, so if this is the case I would suggest the authors instead say that the study represents schools in England only.

We have modified text to make it clear that the study is specific to England.

Second, again the article refers to contact patterns in the UK and the general lack of data on contact patterns of

younger children in the school setting. I can understand only focusing on studies in the UK, however, there is a 2016 study (Guclu et al.) on contact patterns of children in schools in the US that does include younger children to a great extent. I am surprised to not see this article referenced or compared to at all. While the countries are different, school structures and mixing patterns in developed Western countries do share many similarities and would be appropriate to make some comparisons. I would suggest the authors look at this work and see how their expert judgements of pre-COVID contact rates and patterns compare with this study to help understand how their work is applicable beyond the UK/England context.

We are grateful to the reviewer for pointing out this important study which also led us to a couple of other relevant publications that we now cite. The results of Guclu et al are very consistent with our results which give slightly greater daily contacts in pre-COVID times, the difference being attributable to the different methodology. After reading this paper it became clear we did not sufficiently justify our choice of contact definition as a face to face conversation at 5 minutes or more at 1 meter. Our definition is in fact similar to Guclu et al in that they also choose 5 minutes as a "significant contact". We have expanded the explanation of our criterion in text and incorporated the findings of Guclu et al into the paper including comparisons with our data. Another interesting aspect is that we are able to justify the choice of 5 minutes or more for significant contacts *a posteriori* using the contact duration distribution data in Guclu et al. A paragraph on this aspect has been added to the discussion section.

Third, the authors mention that "(T)he Classical Model for SEJ has been deployed in several public health policy applications [9-15] but is not widely used in epidemiology". Can you explain a bit more on why this approach hasn't been widely used in the field? Is there a legitimate concern among epidemiologists in using this approach? Is this a result of lack of familiarity with this approach in epidemiology?

We decided to remove the phrase "but is not widely used in epidemiology". Although we think that this is the case it would need a quite wide-ranging discussion to corroborate the assertion and also respond to the reviewer's justified questions. Such a discussion would be distracting from the theme of our paper and would be better developed in another forum.

It's briefly mentioned that volunteer experts are from the Royal Society Schools network and have STEM backgrounds. I think it would be useful to include a statement on how this might bias the results of the expert opinions. Are these educators and staff from schools considered better than average? Do these schools have fewer children of key workers and/or from vulnerable environments? Do Royal Society Schools network volunteers skew towards higher performing schools or do they come from a broad spectrum of schools in the UK?

We have added some relevant information to address these points related to how representative the schools and teachers are. There is some slight slant towards schools in regions with higher participation in higher education, but this is not expected to introduce any significant distortion of our findings.

I found lines 120-123 a bit confusing. Clarification here would be great. Specifically are you saying that 28 experts completed the calibration questions and 26 of those also answered the 2nd questionnaire? It's not clear to me what was meant here.

We agree that the writing was muddled and have changed and simplified the wording. We have also made some minor adjustments in other places so that the numbers of schools and volunteers are consistent.

The authors also state that the schools of the experts surveyed are nationally representative because the pupil to teacher ratios are in line with national averages. However, they also state that the sizes of the schools of these experts are 30% greater than the national average. I don't think you can say that the schools of the experts surveyed are nationally representative then in general. I would simply ask the language change to be more specific.

Here we have deleted the sentence stating that the schools are nationally representative. The reader can then make this judgement on the basis of the information provided.

Line 176 has a period in red text.

Fixed

In the Methods section there is a value referred to as a 'p-value'. However, this isn't a p-value in the usual sense and I think it would be better to simply refer to it by another name to avoid possible confusion (this is mentioned in the text but changing the name would probably be better).

This is a p-value determined by the goodness-of-fit to the (Chi-squared) distribution implied by the expert's quantile judgements, which may be construed as a hypothesis that the expert is "statistically accurate". Thus, we use the p-

value as a scoring metric (for combining with the expert's information score), and not as a conventional hypothesis test; the reviewer is right in that. Its provenance comes from the theory of proper scoring rules (see Cooke ref 7). We have modified the text here to make this clearer."

Overall, if the issues raised can be addressed I think the article can be recommended for publication. Some of the material could find its way instead to the supplementary material if anything. My last recommendation would be that the discussion ends on a stronger note. In its current form it ends in a limitation rather than driving home the message of the article.

We have added an additional paragraph.

Reviewer: 2

Comments to the Author(s)

I must preface my review by saying that I am a close colleague and personal friend of one of the authors (Cooke), know one of the others (Aspinall), and am involved in an active proposal with them for research funding from the US National Science Foundation.

This paper uses a novel approach to address a problem of great practical significance -- i.e., the efficacy of mitigation approaches intended to reduce contact in primary schools in England during the pandemic.

The method -- formally-elicited structured expert judgment -- is deftly applied and properly interpreted by some of its leading practitioners.

The paper is, in general, quite clearly written. However some sections (e.g., Results -- especially section 3.2 Elicitation Data) are not. In my view, effort to carefully rewrite these sections would increase the impact of the work.

The extent to which contact, as opposed to airborne transmission, is important in controlling the spread of COVID-19, is an open question. But, this paper contributes important information about the extent of contact in schools and the efficacy of mitigation measures in reducing contact.

In my view the paper would be much stronger if the authors focused on storytelling -- what questions does the paper address? what methods does it rely on to answer these? ... and, perhaps most importantly, what are the answers?

As now written, the manuscript seems to get distracted in some sections on technical details -- which are often not clearly explained.

I believe that an effort to rewrite the paper with this in mind would pay off in terms of broad accessibility and ultimate impact.

I wonder whether in the description of the elicitation results it might not be better to carefully explain the results for one cohort (for example, discussing in some detail the interpretation of responses to questions 2a, 2b and 2c) ... and then to relegate the presentation of the detailed responses to similar questions for the other cohorts to a technical appendix.

One technical concern that I have is that uncertainty and variability are not clearly distinguished.

I know that the authors are well aware of the issue. I would urge them to revise the paper to make clear whether the distributions produced in question 2c, 3c, 5c, 6c, 8c, 9c, 10c and 11c -- reflect variability of contact or a mix of variability of and uncertainty about contact

These have been clarified and more details are given below of the revisions.

It was also unclear to me whether the variability is variability across students, days or student-days.

----- NOTE -- Detailed Editorial Comments Below -----

Examples of places where the writing could be improved include:

-- line 95 ... would be nice to know how many, rather than 'most'

Revised as suggested

-- lines 121-123 ... possibly a word or two missing from this sentence

These sentences have been rewritten also in response to reviewer 1.

-- line 154 and Table 1 ... in the elicitation of expert judgment a key issue is whether the elicitation questions satisfy the 'clairvoyance' criterion -- it is not completely clear to me that all of the questions satisfy clairvoyance -- e.g., question 2b -- 'give a range of contacts' -- I'm sure that, as leading experts in the field, the authors have thought carefully about this.

We have referred the reader here more explicitly to the supplementary material where the questionnaire definitions and guidance to the experts is explained. We have added two extra sentences within the supplementary material.

-- lines 257-260 and Fig 2 ... not clear why experts were shown in this particular order on the figure ... possibly better if ordered by performance score

We prefer to keep the order as is. In this exercise, we were not concerned with analysing the experts' performances per se, but in the immediate collective results from the group. Figure 2 has been revised and more fully explained in the caption to make it clear the order has no significance.

-- first few para of section 3.2 are quite dense and opaque

We accept this criticism and have addressed it as follows. We realized that earlier in the paper we had not described, explained and justified the elicitation questionnaire adequately so the reader can be clear about the meaning and purpose of each question when they come to the results section. Thus we have rewritten and expanded section 2.3.1. We have also expanded and re-organised the text in the new section 3.2.1 which integrates the description of the results including characterization of the uncertainties.

-- lines 284-286 ... since these questions are at the heart of the study ... this needs to be explained much more clearly.

The questions are now described more thoroughly in revised and expanded section 2.3.1.

-- lines 286-287 and Fig 3 ... one sentence 'shows examples of responses to illustrate variation in responses among experts' -- seems like an inadequate explanation of six highly detailed graphs -- either drop 5 of the 6 graphs or do a better job of explaining these

We have reduced the number of graphs in Figure 3 from 6 to 2 examples. We have chosen as an example results for one cohort with a graph for pre-COVID and a graph for new normal times. The text now has a fuller explanation of the characteristics of the range graphs.

--lines 287-288 and Fig 4 -- tremendous figure ...key result not clear why the word 'Cases' was used to describe the different cohorts ...also not clear that the 5th, 50th and 95th here refer to variability, rather than uncertainty ... would be worth clarifying.

We have re-written and avoided "Cases" and agree this is a key graph for the study. The reviewer's comment here reflects our original organisation where discussion of Figure 4 was in two different sections, but now we have integrated these. There was already a specific statement about the sources of uncertainty that contributed to the range. Following a paragraph of discussion we write: "In creating the DM we are combining measurement accuracy at schools, real variations in contacts between schools and significantly different calibrations."

--lines 289-291 and Fig 6 -- important figure ... but very busy and quite poorly explained ... almost uninterpretable ... what are the 0, 45 and 75% lines? ... do the 5%, mean, and 95% listed parenthetically on the x-axis label reflect variability in the contact rates within the various cohorts or uncertainty or both?/ ... this would seem to be one of the key results of the study and deserves more careful explanation.

We have expanded the text and modified the caption to improve the explanation, including the meaning of the lines. We interpret the range (5% and 95%) as an overall uncertainty that includes both variability and knowledge uncertainty; one might say epistemic and aleatory uncertainties

-- lines 292-295 ... the meaning of 'The interviews conducted in a smaller sample of experts' is unclear

Rewritten to clarify

-- lines 329-342 ... the methodology here is central to the construction of distribution of individual R_0 values ... but this para leaves some ambiguity about exactly what was done ... since the tails of these contact distributions may be important, the details here matter.

As mentioned above we have addressed this issue by a revised and expanded section 2.3.1. Here is a new sentence in this section:

"Combining the results for the 50th percentile value from questions of the first kind with values of "least" and "more" from questions of the second kind provides a measure of variability of contacts for individuals in a cohort; this represents a quantitative realization with meaning for epidemiological modelling."

We have also added a phrase in this paragraph to make it clearer how these data have been used.

-- lines 356-367 ... make clear whether the unit of observation is the student, the day or the student-day ... if this does not matter for the intended use then make this clear as well.

We have clarified that the estimates are for daily contacts

-- lines 368-381 ... were the people you interviewed aware of this literature? did you ask the subjects to explain the evidence they relied on and why they chose to emphasize certain evidence and give less emphasis to other evidence ... some would argue that understanding the reasoning of the experts is as important as the quantitative results of the elicitation.

The experts and specifically the people interviewed are unlikely to have been aware of the literature. We were conscious of the stress and challenges on the teachers at the time of the two elicitations as they struggled to re-open schools under COVID restrictions. Indeed, we conducted the structured interviews with a view to gaining insights into the reasoning of the responding teachers when they were severely challenged for time. Time constraints on the investigator who conducted the interviews, and the limited number of teachers who subsequently available to be interviewed post hoc meant there was a limit to how much we could infer about their evidential reasoning.

-- lines 598-606 -- it is great that the schools you studied seem in some ways -- 'size and community' -- to be similar in these ways to the broad population of elementary schools in England -- but it is not obvious that they are similar in terms of socioeconomic status of pupils, parents or staff -- all of which could influence the resources available to respond to and the behaviours necessary to adapt to the pandemic.

We have adjusted the wording to tone down the inference that they are fully representative of the country. However we have also added further information about the schools which strengthen our claim. First all but one of the schools are state schools. Second the schools are well distributed across all quintiles of a classification based on the likelihood of children participating in higher education, although there is a slight bias toward higher attaining areas.

-- lines 633-640 -- seem not entirely clear to me

We have simplified the wording

Appendix B

Response to further review and editor comments

We have inserted an extra short paragraph (highlighted in yellow) in the discussion section on the mechanisms of transmission of COVID19 due to aerosols dispersal in enclosed spaces as suggested by the reviewer. We have included three additional references of this topic. Our study does not directly investigate this mechanism, so we are simply acknowledging that this mechanism has been propose and note that at least qualitatively risk from this mechanism also increases with class size and duration.

We have made some very minor corrections of typos, changed the affiliation address of one author and included supporting grants in the acknowledgements.

Steve Sparks

(on behalf of the authors)

12th January 2021